# Diffusion Attribution Score: Evaluating Training Data Influence in Diffusion Models

**Jinxu Lin**[1]  **Linwei Tao**[1]  **Minjing Dong**[2]  **Chang Xu**[1]
[1]The University of Sydney,  [2]City University of Hong Kong
{jinxu.lin, linwei.tao, c.xu}@sydney.edu.au, minjdong@cityu.edu.hk

## Abstract

As diffusion models become increasingly popular, the misuse of copyrighted and private images has emerged as a major concern. One promising solution to mitigate this issue is identifying the contribution of specific training samples in generative models, a process known as data attribution. Existing data attribution methods for diffusion models typically quantify the contribution of a training sample by evaluating the change in diffusion loss when the sample is included or excluded from the training process. However, we argue that the direct usage of diffusion loss cannot represent such a contribution accurately due to the calculation of diffusion loss. Specifically, these approaches measure the divergence between predicted and ground truth distributions, which leads to an indirect comparison between the predicted distributions and cannot represent the variances between model behaviors. To address these issues, we aim to measure the direct comparison between predicted distributions with an attribution score to analyse the training sample importance, which is achieved by Diffusion Attribution Score (*DAS*). Underpinned by rigorous theoretical analysis, we elucidate the effectiveness of DAS. Additionally, we explore strategies to accelerate DAS calculations, facilitating its application to large-scale diffusion models. Our extensive experiments across various datasets and diffusion models demonstrate that DAS significantly surpasses previous benchmarks in terms of the linear data-modelling score, establishing new state-of-the-art performance. [1]

## 1 Introduction

Diffusion models, highlighted in key studies (Ho et al., 2020; Song et al., 2021b), are advancing significantly in generative machine learning with broad applications from image generation to artistic creation (Saharia et al., 2022; Hertz et al., 2023; Li et al., 2022; Ho et al., 2022). As these models, exemplified by projects like Stable Diffusion (Rombach et al., 2022), become increasingly capable of producing high-quality, varied outputs, the misuse of copyrighted and private images has become a significant concern. A key strategy to address this issue is identifying the contributions of training samples in generative models by evaluating their influence on the generation, a task known as data attribution. Data attribution in machine learning is to trace model outputs back to influential training examples, which is essential to understand how specific data points affect model behaviors. In practical applications, data attribution spans various domains, including explaining predictions (Koh & Liang, 2017; Yeh et al., 2018; Ilyas et al., 2022), curating datasets (Khanna et al., 2019; Jia et al., 2021; Liu et al., 2021), and dissecting the mechanisms of generative models like GANs and VAEs (Kong & Chaudhuri, 2021; Terashita et al., 2021), serving to enhance model transparency and explore the effect of training data on model behaviors.

Data attribution methods generally fall into two categories. The first, based on sampling (Shapley et al., 1953; Ghorbani & Zou, 2019; Ilyas et al., 2022), involves retraining models to assess how outputs change with the a data deletion. While effective, this method requires training thousands of models. The second approach uses approximations to assess the change in output function for efficiency (Koh & Liang, 2017; Feldman & Zhang, 2020; Pruthi et al., 2020) by proposing attribution score function. TRAK (Park et al., 2023) introduced a innovative estimator, which considers

---

[1]Code is available at https://github.com/Jinxu-Lin/DAS.

the inverse-sigmoid function as model output and computes attribution scores to assess its change. Building on this, Zheng et al. (2024) proposed D-TRAK, adapting TRAK to diffusion models by following TRAK's derivation and replacing the inverse-sigmoid function with the diffusion loss. Moreover, D-TRAK reported counterintuitive findings that the output function could be replaced with alternative functions without altering the score's form. These empirical designs outperformed theoretically motivated diffusion losses in experiments, emphasizing the need for a deeper understanding of the attribution properties on diffusion models.

In this paper, we define the objective of data attribution in diffusion models as evaluating the impact of training samples on generation by measuring shifts in the predicted distribution after removing specific samples and retraining the model. From this perspective, directly applying TRAK to diffusion models by replacing the output function with the diffusion loss leads to indirect comparisons between predicted distributions, as the diffusion loss represents the KL-divergence between predicted and ground truth distributions (Ho et al., 2020). We also address the counter-intuitive findings in D-TRAK, which manually removes the effect of data distributions in attribution but treats the diffusion model's output as a scalar, failing to capture the unique characteristics of diffusion models. Our analysis indicates that TRAK's derivation cannot be directly extended to diffusion models, as they fundamentally differ from discriminative models. To address this, we propose the Diffusion Attribution Score (*DAS*), a novel metric designed specifically for diffusion models to quantify the impact of training samples by measuring the KL-divergence between predicted distributions when a sample is included or excluded from the training set. DAS computes this divergence through changes in the noise predictor's output, which, by linearization, can be represented as variations in the model parameters. These parameter changes are then measured using Newton's Method. Since directly computing the full form of DAS is computationally expensive, we propose several techniques, such as compressing models and datasets, to accelerate computation. These enhancements enable the application of DAS to large-scale diffusion models, significantly improving its practicality. Our experiments across ranges of datasets and diffusion models demonstrate that DAS significantly outperforms previous benchmarks, including D-TRAK and TRAK, achieving superior results in terms of the linear data-modeling score. This consistent performance across diverse settings highlights its robustness and establishes DAS as the new state-of-the-art method for data attribution in diffusion models. The primary contributions of our work are summarized as follows:

1. We provide a comprehensive analysis of the limitations of directly applying TRAK to diffusion models and evaluate D-TRAK's empirical design, highlighting the need for more effective attribution methods tailored to diffusion models.

2. We introduce DAS, a theoretically solid metric designed to directly quantify discrepancies in model outputs, supported by detailed derivations. We also discuss various techniques, such as compressing models or datasets, to accelerate the computation of DAS, facilitating its efficient implementation.

3. DAS demonstrates state-of-the-art performance across multiple benchmarks, notably excelling in linear datamodeling scores.

## 2 RELATED WORKS

### 2.1 DATA ATTRIBUTION

The training data exerts a significant influence on the behavior of machine learning models. Data attribution aims to accurately assess the importance of each piece of training data in relation to the desired model outputs. However, methods of data attribution often face the challenge of balancing computational efficiency with accuracy. Sampling-based approaches, such as empirical influence functions (Feldman & Zhang, 2020), Shapley value estimators (Ghorbani & Zou, 2019; Jia et al., 2019), and datamodels (Ilyas et al., 2022), are able to precisely attribute influences to training data but typically necessitate the training of thousands of models to yield dependable results. On the other hand, methods like influence approximation (Koh & Liang, 2017; Schioppa et al., 2022) and gradient agreement scoring (Pruthi et al., 2020) provide computational benefits but may falter in terms of reliability in non-convex settings (Basu et al., 2021; Akyurek et al., 2022).

## 2.2 DATA ATTRIBUTION IN GENERATIVE MODELS

The discussed methods address counterfactual questions within the context of discriminative models, focusing primarily on accuracy and model predictions. Extending these methodologies to generative models presents complexities due to the lack of clear labels or definitive ground truth. Research in this area includes efforts to compute influence within Generative Adversarial Networks (GANs) (Terashita et al., 2021) and Variational Autoencoders (VAEs) (Kong & Chaudhuri, 2021). In the realm of diffusion models, earlier research (Dai & Gifford, 2023) has explored influence computation by employing ensembles that necessitate training multiple models on varied subsets of training data—a method less suited for traditionally trained models. Wang et al. (2023) suggest an alternative termed "customization," which involves adapting or tuning a pretrained text-to-image model through a specially designed training procedure. MONTAGE (Brokman et al., 2025) integrates a novel technique to monitor generations throughout the training via internal model representations. In this paper, we mainly focus on post-hoc data attribution method, which entails applying attribution methods after training. Recently, Park et al. (2023) developed TRAK, a new attribution method that is both effective and computationally feasible for large-scale models. Journey TRAK (Georgiev et al., 2023) extends TRAK to diffusion models by attributing influence across individual denoising timesteps. Moreover, D-TRAK (Zheng et al., 2024) has revealed surprising results, indicating that theoretically dubious choices in the design of TRAK might enhance performance, highlighting the imperative for further exploration into data attribution within diffusion models. In DataInf (Kwon et al., 2024), influence function using the loss gradient and Hessian have been improved for greater accuracy and efficiency in attributing diffusion models. A concurrent work (Mlodozeniec et al., 2025) extends influence functions to diffusion models by predicting how the probability of generating specific examples changes when training data is removed, leveraging a K-FAC approximation for scalable computation. These studies are pivotal in advancing our understanding and fostering the development of instance-based interpretations in unsupervised learning contexts. We give a more detailed theoretical discussion about the existing data attribution methods in diffusion model in Appendix E.3. Besides, we provide an introduction about application of data attribution in Appendix I.

## 3 PRELIMINARIES

### 3.1 DIFFUSION MODELS

Our study concentrates on discrete-time diffusion models, specifically Denoising Diffusion Probabilistic Models (DDPMs) (Ho et al., 2020) and Latent Diffusion Models (LDMs) which are foundational to Stable Diffusion (Rombach et al., 2022). This paper grounds all theoretical derivations within the framework of unconditional generation using DDPMs. Below, we detail the notation employed in DDPMs that underpins all further theoretical discussions.

Consider a training set $\mathbb{S} = \{\boldsymbol{z}^{(1)}, ..., \boldsymbol{z}^{(n)}\}$ where each training sample $\boldsymbol{z}^{(i)} := (\boldsymbol{x}^{(i)}, y^{(i)}) \sim \mathcal{Z}$ is an input-label pair[2]. Given an input data distribution $q(\boldsymbol{x})$, DDPMs aim to model a distribution $p_\theta(\boldsymbol{x})$ to approximate $q(\boldsymbol{x})$. The learning process is divided into forward and reverse process, conducted over a series of timesteps in the latent variable space, with $\boldsymbol{x}_0$ denoting the initial image and $\boldsymbol{x}_t$ the latent variables at timestep $t \in [1, T]$. In the forward process, DDPMs sample an observation $\boldsymbol{x}_0$ from $\mathbb{S}$ and add noise on it across $T$ timesteps: $q(\boldsymbol{x}_t|\boldsymbol{x}_{t-1}) := \mathcal{N}(\boldsymbol{x}_t; \sqrt{1 - \beta_t}\boldsymbol{x}_{t-1}, \beta_t \boldsymbol{I})$, where $\beta_1, ..., \beta_T$ constitute a variance schedule. As indicated in DDPMs, the latent variable $\boldsymbol{x}_t$ can be express as a linear combination of $\boldsymbol{x}_0$:

$$\boldsymbol{x}_t = \sqrt{\bar{\alpha}_t}\boldsymbol{x}_0 + \sqrt{1 - \bar{\alpha}_t}\boldsymbol{\epsilon}, \tag{1}$$

where $\alpha_t := 1 - \beta_t, \bar{\alpha}_t := \prod_{s=1}^{t} \alpha_s$ and $\boldsymbol{\epsilon} \sim \mathcal{N}(0, \boldsymbol{I})$. In the reward process, DDPMs model a distribution $p_\theta(\boldsymbol{x}_{t-1}|\boldsymbol{x}_t)$ by minimizing the KL-divergence from data at $t$:

$$D_{\mathrm{KL}}[p_\theta(\boldsymbol{x}_{t-1}|\boldsymbol{x}_t)\|q(\boldsymbol{x}_{t-1}|\boldsymbol{x}_t, \boldsymbol{x}_0)] = \mathbb{E}_{\boldsymbol{\epsilon}\sim\mathcal{N}(0,\boldsymbol{I})}[\frac{\beta_t^2}{2\alpha_t(1 - \bar{\alpha}_t)}\|\boldsymbol{\epsilon} - \boldsymbol{\epsilon}_\theta(\boldsymbol{x}_t, t)\|^2], \tag{2}$$

---

[2]In text-to-image task, $\boldsymbol{z} := (\boldsymbol{x}^i, y^i)$ represents an image-caption sample, whereas in unconditional generation, it solely contains an image $\boldsymbol{z} := (\boldsymbol{x})$.

where $\epsilon_\theta$ is a function implemented by models $\theta$ which can be seen as a noise predictor. A simplified version of objective function for a data point $\boldsymbol{x}$ used to train DDPMs is:

$$\mathcal{L}_{\text{Simple}}(\boldsymbol{x}, \theta) = \mathbb{E}_{\epsilon, t}[||\epsilon_\theta(\boldsymbol{x}_t, t) - \epsilon||^2]. \tag{3}$$

## 3.2 DATA ATTRIBUTION

Given a DDPM trained on dataset $\mathbb{S}$, our objective is to trace the influence of the training data on the generation of sample $\boldsymbol{z}$. This task is referred to as data attribution, which is commonly solved by addressing a counterfactual question: removing a training sample $\boldsymbol{z}^{(i)}$ from $\mathbb{S}$ and retraining a model $\theta_{\backslash i}$ on the subset $\mathbb{S}_{\backslash i}$, the influence of $\boldsymbol{z}^{(i)}$ on $\boldsymbol{z}$ can be assessed by the change in the model output, computed as $f(\boldsymbol{z}, \theta) - f(\boldsymbol{z}, \theta_{\backslash i})$. The function $f(\boldsymbol{z}, \theta)$, which represents the model output, has a variety of choices, such as the direct output of the model or the loss function.

To avoid the high costs of model retraining, some data attribution methods compute a score function $\tau(\boldsymbol{z}, \mathbb{S}) : \mathcal{Z} \times \mathcal{Z}^n \to \mathbb{R}^n$ to reflect the importance of each training sample in $\mathbb{S}$ on the sample $\boldsymbol{z}$. For clarity, $\tau(\boldsymbol{z}, \mathbb{S})^{(i)}$ denotes the attribution score assigned to the individual training sample $\boldsymbol{z}^{(i)}$ on $\boldsymbol{z}$. TRAK (Park et al., 2023) stands out as a representative data attribution method designed for large-scale models focused on discriminative tasks. TRAK defines the model output function as:

$$f_{\text{TRAK}}(\boldsymbol{z}, \theta) = \log[\hat{p}(\boldsymbol{x}, \theta)/(1 - \hat{p}(\boldsymbol{x}, \theta))], \tag{4}$$

where $\hat{p}(\boldsymbol{x}, \theta)$ is the corresponding class probability of $\boldsymbol{z}$. TRAK then introduces attribution score $\tau_{\text{TRAK}}(\boldsymbol{z}, \mathbb{S})^{(i)}$ to approximate the change in $f_{\text{TRAK}}$ after a data intervention, which is expressed as:

$$\tau_{\text{TRAK}}(\boldsymbol{z}, \mathbb{S})^{(i)} := \phi(\boldsymbol{z})^\top (\boldsymbol{\Phi}^\top \boldsymbol{\Phi})^{-1} \phi(\boldsymbol{z}^{(i)}) r^{(i)} \approx f_{\text{TRAK}}(\boldsymbol{z}, \theta) - f_{\text{TRAK}}(\boldsymbol{z}, \theta_{\backslash i}), \tag{5}$$

where $r^{(i)} := [1 - \hat{p}(\boldsymbol{x}^{(i)}, \theta)]$ denotes the residual for sample $\boldsymbol{z}^{(i)}$. Here, $\phi(\boldsymbol{z}) := \boldsymbol{P}^\top \nabla_\theta f_{\text{TRAK}}(\boldsymbol{z}, \theta)$ and $\boldsymbol{\Phi} := [\phi(\boldsymbol{z}^{(1)}), ..., \phi(\boldsymbol{z}^{(n)})]$ represents the matrix of stacked gradients on $\mathbb{S}$. $\boldsymbol{P} \sim \mathcal{N}(0, 1)^{d \times k}$ is a random projection matrix (Johnson & Lindenstrauss, 1984) to reduce the gradient dimension.

To adapt TRAK in diffusion models, D-TRAK (Zheng et al., 2024) first followed TRAK's guidance, replacing the output function with Simple Loss: $f_{\text{D-TRAK}}(\boldsymbol{z}, \theta) = \mathcal{L}_{\text{Simple}}(\boldsymbol{x}, \theta)$, and simplifies the residual term to an identity matrix $\boldsymbol{I}$. The attribution function in D-TRAK is defined as follows:

$$\tau_{\text{D-TRAK}}(\boldsymbol{z}, \mathbb{S})^{(i)} = \phi(\boldsymbol{z})^\top (\boldsymbol{\Phi}^\top \boldsymbol{\Phi})^{-1} \phi(\boldsymbol{z}^{(i)}) \boldsymbol{I} \approx f_{\text{D-TRAK}}(\boldsymbol{z}, \theta) - f_{\text{D-TRAK}}(\boldsymbol{z}, \theta_{\backslash i}), \tag{6}$$

where $\phi(\boldsymbol{z}) := \boldsymbol{P}^\top \nabla_\theta f_{\text{D-TRAK}}(\boldsymbol{z}, \theta)$. Interestingly, D-TRAK observed that substituting Simple Loss with other functions can yield superior attribution performance. Examples of these include $\mathcal{L}_{\text{Square}}(\boldsymbol{z}, \theta) = \mathbb{E}_{t, \epsilon}[||\epsilon_\theta(\boldsymbol{x}_t, t)||^2]$ and $\mathcal{L}_{\text{Average}}(\boldsymbol{z}, \theta) = \mathbb{E}_{t, \epsilon}[\text{Avg}(\epsilon_\theta(\boldsymbol{x}_t, t))]$.

## 4 METHODOLOGY

### 4.1 RETHINKING OUTPUT FUNCTION IN DIFFUSION MODELS

For the goal of data attribution within diffusion models, we want to measure the influence of a specific training sample $\boldsymbol{z}^{(i)}$ on a generated sample $\boldsymbol{z}^{\text{gen}}$. This task can be approached by addressing the counterfactual question: How would $\boldsymbol{z}^{\text{gen}}$ change if we removed $\boldsymbol{z}^{(i)}$ from $\mathbb{S}$ and retrained the model on the subset $\mathbb{S}_{\backslash i}$? This change can be evaluated by the distance between predicted distribution $p_\theta(\boldsymbol{x}^{\text{gen}})$ and $p_{\theta_{\backslash i}}(\boldsymbol{x}^{\text{gen}})$ in DDPM, where the subscript denotes the deletion of $\boldsymbol{z}^{(i)}$.

Reviewing D-TRAK, they proposes using $\tau_{\text{D-TRAK}}$ to approximate this difference on the Simple Loss by setting output function as $f_{\text{D-TRAK}} = \mathcal{L}_{\text{Simple}}$. In details, the difference is computed as:

$$\tau_{\text{D-TRAK}}(\boldsymbol{z}^{\text{gen}}, \mathbb{S})^{(i)} \approx f_{\text{D-TRAK}}(\boldsymbol{z}^{\text{gen}}, \theta) - f_{\text{D-TRAK}}(\boldsymbol{z}^{\text{gen}}, \theta_{\backslash i})$$
$$= \mathbb{E}_{\epsilon, t}[||\epsilon_\theta(\boldsymbol{x}_t^{\text{gen}}, t) - \epsilon||^2] - \mathbb{E}_{\epsilon, t}[||\epsilon_{\theta_{\backslash i}}(\boldsymbol{x}_t^{\text{gen}}, t) - \epsilon||^2]. \tag{7}$$

However, from the distribution perspective, this approach conducts an indirect comparison of KL-divergences between the predicted distributions by involving the data distribution $q(\boldsymbol{x}^{\text{gen}})$:

$$\tau_{\text{D-TRAK}}(\boldsymbol{z}^{\text{gen}}, \mathbb{S})^{(i)} \approx D_{\text{KL}}[p_\theta(\boldsymbol{x}^{\text{gen}})||q(\boldsymbol{x}^{\text{gen}})] - D_{\text{KL}}[p_{\theta_{\backslash i}}(\boldsymbol{x}^{\text{gen}})||q(\boldsymbol{x}^{\text{gen}})]$$
$$= D_{\text{KL}}[p_\theta(\boldsymbol{x}^{\text{gen}})||p_{\theta_{\backslash i}}(\boldsymbol{x}^{\text{gen}})] - \int [p_\theta(\boldsymbol{x}^{\text{gen}}) - p_{\theta_{\backslash i}}(\boldsymbol{x}^{\text{gen}})] \log q(\boldsymbol{x}^{\text{gen}}) \mathrm{d}x. \tag{8}$$

Compared to directly computing the KL-divergences between predicted distributions $p_\theta$ and $p_{\theta_{\setminus i}}$, D-TRAK involves a Cross Entropy between the predicted distribution shift $p_\theta - p_{\theta_{\setminus i}}$ and data distribution $q$, where the crossing term is generally nonzero unless are identical distributions. This setting involves the influence of the data distribution $q$ in the attribution process and may introduce errors when evaluating the differences. We can consider the irrationality of this setting through a practical example: the predicted distributions might approach the data distribution $q(\boldsymbol{x}^{\text{gen}})$ from different directions while training, yet exhibit similar distances. In this case, D-TRAK captures only minimal loss changes, failing to reflect the true distance between the predicted distributions.

To isolate the effect of the removed data on the model, we propose the Diffusion Attribution Score (DAS), conducting a direct comparison between $p_\theta$ and $p_{\theta_{\setminus i}}$ by assessing their KL-divergence:

$$
\begin{aligned}
\tau_{\text{DAS}}(\boldsymbol{z}^{\text{gen}}, \mathbb{S})^{(i)} &\approx D_{\text{KL}}[p_\theta(\boldsymbol{x}^{\text{gen}}) || p_{\theta_{\setminus i}}(\boldsymbol{x}^{\text{gen}})] \\
&\approx \mathbb{E}_{\boldsymbol{\epsilon}, t}[||\boldsymbol{\epsilon}_\theta(\boldsymbol{x}_t^{\text{gen}}, t) - \boldsymbol{\epsilon}_{\theta_{\setminus i}}(\boldsymbol{x}_t^{\text{gen}}, t)||^2].
\end{aligned}
\tag{9}
$$

The output function in DAS is defined as $f_{\text{DAS}}(\boldsymbol{z}, \theta) = \boldsymbol{\epsilon}_\theta(\boldsymbol{x}_t^{\text{gen}}, t)$, which is able to directly reflect the differences between the noise predictors of the original and the retrained models. Eq. 9 also validates the effectiveness of employing $\mathcal{L}_{\text{square}}$ as the output function, which is formulated as:

$$
\tau_{\text{Square}}(\boldsymbol{z}^{\text{gen}}, \mathbb{S})^{(i)} \approx \mathbb{E}_{\boldsymbol{\epsilon}, t}[||\boldsymbol{\epsilon}_\theta(\boldsymbol{x}_t^{\text{gen}}, t)||^2] - \mathbb{E}_{\boldsymbol{\epsilon}, t}[||\boldsymbol{\epsilon}_{\theta_{\setminus i}}(\boldsymbol{x}_t^{\text{gen}}, t)||^2].
\tag{10}
$$

The effectiveness of $\tau_{\text{Square}}$ lies in manually eliminating the influence of $q$; however, this approach has its limitations since the output of diffusion model is a high dimensional feature. Defining the output function as $\mathcal{L}_{\text{Square}}$ or using average and $L^2$ norm, treats the latent as a scalar, thereby neglecting dimensional information. For instance, these matrices might exhibit identical differences across various dimensions, an aspect that scalar representations fail to capture. This dimensional consistency is crucial for understanding the full impact of training data alterations on model outputs.

## 4.2 DIFFUSION ATTRIBUTION SCORE

Since we define the output change as KL-divergence, which differs from TRAK, $f_{\text{DAS}}$ cannot be directly applied to TRAK and $\tau_{\text{DAS}}$ needs to be specifically derived on diffusion models. In this section, we explore methods to approximate Eq. 9 at timestep $t$ without retraining the model. The derivation is divided into two main parts: First, we linearize the output function, allowing the difference in the output function to be expressed in terms of difference in model parameters. The second part is that approximating this relationship using Newton's method. By integrating these two components, we derive the complete formulation of DAS to attribute the output of diffusion model at timestep $t$.

**Linearizing Output Function.** Computing the output of the retrained model $\boldsymbol{\epsilon}_{\theta_{\setminus i}}(\boldsymbol{x}_t^{\text{gen}}, t)$ is computationally expensive. For computational efficiency, we propose linearizing the model output function around the optimal model parameters $\theta^*$ at convergence, simplifying the calculation as follows:

$$
f_{\text{DAS}}(\boldsymbol{z}_t, \theta) \approx \boldsymbol{\epsilon}_{\theta^*}(\boldsymbol{x}_t, t) + \nabla_\theta \boldsymbol{\epsilon}_{\theta^*}(\boldsymbol{x}_t, t)^\top (\theta - \theta^*).
\tag{11}
$$

By substituting Eq. 11 into Eq. 9, we derive:

$$
\tau_{\text{DAS}}(\boldsymbol{z}^{\text{gen}}, \mathbb{S})_t^{(i)} \approx \mathbb{E}_{\boldsymbol{\epsilon}}[||\nabla_\theta \boldsymbol{\epsilon}_{\theta^*}(\boldsymbol{x}_t^{\text{gen}}, t)^\top (\theta^* - \theta_{\setminus i}^*)||^2].
\tag{12}
$$

The subscript $t$ indicates the attribution for the model output at timestep $t$. Consequently, the influence of removing a sample can be quantitatively evaluated through the changes in model parameters, which can be measured by the Newton's method, thereby reducing the computational overhead.

**Estimating the model parameter.** Consider using the leave-one-out method, the variation of the model parameters can be assessed by Newton's Method (Pregibon, 1981). The counterfactual parameters $\theta_{\setminus i}^*$ can be approximated by taking a single Newton step from the optimal parameters $\theta^*$:

$$
\theta^* - \theta_{\setminus i}^* \leftarrow -[\nabla_\theta \boldsymbol{\epsilon}_{\theta^*}(\mathbb{S}_{\setminus i_t}, t)^\top \nabla_\theta \boldsymbol{\epsilon}_{\theta^*}(\mathbb{S}_{\setminus i_t}, t)]^{-1} \nabla_\theta \boldsymbol{\epsilon}_{\theta^*}(\mathbb{S}_{\setminus i}, t)^\top \boldsymbol{R}_{\setminus i_t},
\tag{13}
$$

where $\boldsymbol{\epsilon}_{\theta^*}(\mathbb{S}_t, t) := [\boldsymbol{\epsilon}_{\theta^*}(\boldsymbol{x}_t^{(1)}, t), ..., \boldsymbol{\epsilon}_{\theta^*}(\boldsymbol{x}_t^{(n)}, t)]$ represents the stacked output matrix for the set $\mathbb{S}$ at timestep $t$, and $\boldsymbol{R}_t := \text{diag}[\boldsymbol{\epsilon}_\theta(\boldsymbol{x}_t^{(i)}, t) - \boldsymbol{\epsilon}]$ is a diagonal matrix describing the residuals among $\mathbb{S}$. A detailed proof of Eq. 13 is provided in Appendix A.

Let $\boldsymbol{g}_t(\boldsymbol{x}^{(i)}) = \nabla_\theta \boldsymbol{\epsilon}_{\theta^*}(\boldsymbol{x}_t^{(i)}, t)$ and $\boldsymbol{G}_t(\mathbb{S}) = \nabla_\theta \boldsymbol{\epsilon}_{\theta^*}(\mathbb{S}_t, t)$. The inverse term in Eq. 13 can be reformulated as:

$$\boldsymbol{G}_t(\mathbb{S}_{\backslash i})^\top \boldsymbol{G}_t(\mathbb{S}_{\backslash i}) = \boldsymbol{G}_t(\mathbb{S})^\top \boldsymbol{G}_t(\mathbb{S}) - \boldsymbol{g}_t(\boldsymbol{x}^{(i)})^\top \boldsymbol{g}_t(\boldsymbol{x}^{(i)}). \tag{14}$$

Applying the Sherman–Morrison formula to Eq. 14 simplifies Eq. 13 as follows:

$$\theta^* - \theta^*_{\backslash i} \leftarrow \frac{[\boldsymbol{G}_t(\mathbb{S})^\top \boldsymbol{G}_t(\mathbb{S})]^{-1} \boldsymbol{g}_t(\boldsymbol{x}^{(i)}) \boldsymbol{r}_t^{(i)}}{1 - \boldsymbol{g}_t(\boldsymbol{x}^{(i)})^\top [(\boldsymbol{G}_t(\mathbb{S})^\top \boldsymbol{G}_t(\mathbb{S})]^{-1} \boldsymbol{g}_t(\boldsymbol{x}^{(i)})}, \tag{15}$$

where $\boldsymbol{r}_t^{(i)}$ is the $i$-th element of $\boldsymbol{R}_t$. A detailed proof of Eq 15 in provided in Appendix B.

**Diffusion Attribution Score.** By substituting Eq.15 into Eq.12, we derive the formula for computing the DAS at timestep $t$:

$$\tau_{\text{DAS}}(\boldsymbol{z}^{\text{gen}}, \mathbb{S})_t^{(i)} = \mathbb{E}_{\boldsymbol{\epsilon}}[\|\frac{\boldsymbol{g}_t(\boldsymbol{x}^{\text{gen}})[\boldsymbol{G}_t(\mathbb{S})^\top \boldsymbol{G}_t(\mathbb{S})]^{-1} \boldsymbol{g}_t(\boldsymbol{x}^{(i)}) \boldsymbol{r}_t^{(i)}}{1 - \boldsymbol{g}_t(\boldsymbol{x}^{(i)})^\top [(\boldsymbol{G}_t(\mathbb{S})^\top \boldsymbol{G}_t(\mathbb{S})]^{-1} \boldsymbol{g}_t(\boldsymbol{x}^{(i)})}\|^2]. \tag{16}$$

This equation estimates the impact of training samples at a specific timestep $t$. The overall influence of a training sample $\boldsymbol{z}^{(i)}$ on the target sample $\boldsymbol{z}^{\text{gen}}$ throughout the entire generation process can be computed as an expectation over timestep $t$. However, directly calculating these expectations is extremely costly. In the next section, we discuss methods to expedite this computation.

## 4.3 EXTEND DAS TO LARGE-SCALE DIFFUSION MODEL

Calculating Eq. 16 for large-scale diffusion models poses several challenges. The computation of the inverse term is extremely expensive due to the high dimensionality of the parameters. Additionally, gradients must be calculated for all training samples in $\mathbb{S}$, further increasing computational demands. In this subsection, we explore techniques to accelerate the calculation of Eq. 16. These methods can be broadly categorized into two approaches: The first focuses on reducing the gradient computation by minimizing the number of expectations and candidate training samples. The second aims to accelerate the computation of the inverse term by reducing the dimensionality of the gradients.

**Reducing Calculation of Expectations.** Computing $t$ times the equation specified in Eq. 16 is highly resource-intensive due to the necessity of calculating inverse terms. To simplify, we use the average gradient $\overline{\boldsymbol{g}}(\boldsymbol{x})$ and average residual $\overline{\boldsymbol{r}}$ over entire generation, enabling a single computation of Eq.16 to assess overall influence. However, during averaging, these terms may exhibit varying magnitudes across different timesteps, potentially leading to the loss of significant information. To address this, we normalize the gradients and residuals over the entire generation before averaging:

$$\overline{\boldsymbol{g}}(\boldsymbol{x}^{(i)}) = \frac{1}{T} \sum_t \frac{\boldsymbol{g}_t(\boldsymbol{x}^{(i)})}{\sqrt{\sum_{j=1}^T [\boldsymbol{g}_j(\boldsymbol{x}^{(i)})]^2}}, \quad \overline{\boldsymbol{r}}^{(i)} = \frac{1}{T} \sum_t \frac{\boldsymbol{r}_t^{(i)}}{\sqrt{\sum_{j=1}^T [\boldsymbol{r}_j^{(i)}]^2}}. \tag{17}$$

Thus, to attribute the influence of a training sample $\boldsymbol{z}^{(i)}$ on a generated sample $\boldsymbol{z}^{\text{gen}}$ throughout the entire generation process, we redefine Eq. 16 as follows:

$$\tau_{\text{DAS}}(\boldsymbol{z}^{\text{gen}}, \mathbb{S})^{(i)} = \|\frac{\overline{\boldsymbol{g}}(\boldsymbol{x}^{\text{gen}})^\top [\overline{\boldsymbol{G}}(\mathbb{S})^\top \overline{\boldsymbol{G}}(\mathbb{S})]^{-1} \overline{\boldsymbol{g}}(\boldsymbol{x}^{(i)}) \overline{\boldsymbol{r}}^{(i)}}{1 - \overline{\boldsymbol{g}}(\boldsymbol{x}^{(i)})^\top [\overline{\boldsymbol{G}}(\mathbb{S})^\top \overline{\boldsymbol{G}}(\mathbb{S})]^{-1} \overline{\boldsymbol{g}}(\boldsymbol{x}^{(i)})}\|^2. \tag{18}$$

**Reducing Dimension of Gradients by Projection.** The dimension of $\boldsymbol{g}_t(\boldsymbol{x}^{(i)})$ matches that of the amount of diffusion model's parameter, posing a challenge in calculating the inverse term due to its substantial size. One effective method of reducing the dimension is to apply the Johnson and Lindenstrauss Projection (Johnson & Lindenstrauss, 1984). It involves multiplying the gradient vector $\boldsymbol{g}_t(\boldsymbol{x}^{(i)}) \in \mathbb{R}^p$ by a random matrix $\boldsymbol{P} \sim \mathcal{N}(0, 1) \in \mathbb{R}^{p \times k} (k \ll p)$, which can preserve inner product with high probability while significantly reducing the dimension of the gradient. This projection method has been validated in previous studies (Malladi et al., 2023; Zheng et al., 2024), demonstrating its efficacy in maintaining the integrity of the gradients while easing computational demands. We summarize our algorithms in Algorithm 1 with normalization and projection.

**Reducing Dimension of Gradients by Model Compression** In addition to projection methods, other techniques can be employed to reduce the dimension of gradients in diffusion models. For

instance, as noted by Ma et al. (2024), the up-block of the U-Net architecture in diffusion models plays a pivotal role in the generation process. Therefore, we can focus on the up-block gradients for dimension reduction purposes, optimizing computational efficiency. Furthermore, various strategies have been proposed to fine-tune large-scale diffusion models efficiently. One such approach is LoRA (Hu et al., 2022), which involves freezing the pre-trained model weights while utilizing trainable rank decomposition matrices. This significantly reduces the number of trainable parameters required for fine-tuning. Consequently, when attributing the influence of training samples in a fine-tuned dataset, we can compute the DAS with gradients on the trainable parameters.

**Reducing the amount of timesteps.** Computing Eq. 17 requires performing back propagation $T$ times, making it highly resource-intensive. Sampling fewer timesteps can also approximate the expectation and estimate gradient behavior while significantly lowering computational overhead.

**Reducing Candidate Training Sample.** The necessity to traverse the entire training set when computing the DAS poses a significant challenge. To alleviate this, a practical approach involves conducting a preliminary screening to identify the most influential training samples. Techniques such as CLIP (Radford et al., 2021) or cosine similarity can be effectively employed to locate samples that are similar to the target. By using these methods, we can form a preliminary candidate set and concentrate DAS computations on this subset, rather than on the entire training dataset.

## 5 EXPERIMENTS

### 5.1 DATASETS AND MODELS

In this section, we present a comparative analysis of our method, Diffusion Attribution Score (DAS), against existing data attribution methods across various experimental settings. Our findings demonstrate that DAS significantly outperforms other methods in attribution performance, validating its ability to accurately identify influential training samples. We provide an overview of the datasets and diffusion models used in our experiments, with detailed descriptions available in Appendix E.1.

**CIFAR10 (32×32).** We conduct experiments on the CIFAR-10 (Krizhevsky, 2009), containing 50,000 training samples across 10 classes. For computational efficiency on ablation studies, CIFAR-2, a subset with 5,000 samples randomly selected from 2 classes is proposed. We use a 35.7M DDPM (Ho et al., 2020) with a 50-step DDIM solver (Song et al., 2021a) on these settings.

**CelebA (64×64).** From original CelebA dataset training and test sets (Liu et al., 2015), we extracted 5,000 training samples. Following preprocessing steps outlined by Song et al. (2021b), images were initially center cropped to 140x140 and then resized to 64x64. The diffusion model used mirrors the CIFAR-10 setup but includes an expanded U-Net architecture with 118.8 million parameters.

**ArtBench (256×256).** ArtBench (Liao et al., 2022) is a dataset of 50,000 images across 10 artistic styles. For our studies, we derived two subsets: ArtBench-2, with 5,000 samples from 2 classes and ArtBench-5, with 12,500 samples from five styles. We fine-tuned a Stable Diffusion model on these datasets using LoRA (Hu et al., 2022) with 128 rank and 25.5M parameters. During inference, images are generated using a DDIM solver with a classifier-free guidance (Ho & Salimans, 2021).

### 5.2 EVALUATION METHOD FOR DATA ATTRIBUTION

Various methods are available for evaluating data attribution techniques, including the leave-one-out influence method (Koh & Liang, 2017; Basu et al., 2021) and Shapley values (Lundberg & Lee, 2017). In this paper, we use the Linear Datamodeling Score (LDS) (Ilyas et al., 2022) to assess data attribution methods. Given a model trained $\theta$ on dataset $\mathbb{S}$, LDS evaluates the effectiveness of a data attribution method $\tau$ by initially sampling a sub-dataset $\mathbb{S}' \subset \mathbb{S}$ and retraining a model $\theta'$ on $\mathbb{S}'$. The attribution-based output prediction for an interested sample $\boldsymbol{z}^{\text{test}}$ is then calculated as:

$$g_\tau(\boldsymbol{z}^{\text{test}}, \mathbb{S}', \mathbb{S}) := \sum_{\boldsymbol{z}^{(i)} \in \mathbb{S}'} \tau(\boldsymbol{z}^{\text{test}}, \mathbb{S})^{(i)} \tag{19}$$

The underlying premise of LDS is that the predicted output $g_\tau(\boldsymbol{z}, \mathbb{S}', \mathbb{S})$ should correspond closely to the actual model output $f(\boldsymbol{z}^{\text{test}}, \theta')$. To validate this, LDS samples $M$ subsets of fixed size and

Table 1: LDS (%) on CIFAR-2/CIFAR-10 with timesteps (10 or 100).

| Method | CIFAR2 | | | | CIFAR10 | | | |
|---|---|---|---|---|---|---|---|---|
| | Validation | | Generation | | Validation | | Generation | |
| | 10 | 100 | 10 | 100 | 10 | 100 | 10 | 100 |
| Raw pixel (dot prod.) | 7.77±0.57 | | 4.89±0.58 | | 2.50±0.42 | | 2.25±0.39 | |
| Raw pixel (cosine) | 7.87±0.57 | | 5.44±0.57 | | 2.71±0.41 | | 2.61±0.38 | |
| CLIP similarity (dot prod.) | 6.51±1.06 | | 3.00±0.95 | | 2.39±0.41 | | 1.11±0.47 | |
| CLIP similarity (cosine) | 8.54±1.01 | | 4.01±0.85 | | 3.39±0.38 | | 1.69±0.49 | |
| Gradient (dot prod.) | 5.14±0.60 | 5.07±0.55 | 2.80±0.55 | 4.03±0.51 | 0.79±0.43 | 1.40±0.42 | 0.74±0.45 | 1.85±0.54 |
| Gradient (cosine) | 5.08±0.59 | 4.89±0.50 | 2.78±0.54 | 3.92±0.49 | 0.66±0.43 | 1.24±0.41 | 0.58±0.42 | 1.82±0.51 |
| TracInCP | 6.26±0.84 | 5.47±0.87 | 3.76±0.61 | 3.70±0.66 | 0.98±0.44 | 1.26±0.38 | 0.96±0.40 | 1.39±0.54 |
| GAS | 5.78±0.82 | 5.15±0.87 | 3.34±0.56 | 3.30±0.68 | 0.89±0.48 | 1.25±0.38 | 0.90±0.41 | 1.61±0.54 |
| Journey TRAK | / | / | 7.73±0.65 | 12.21±0.46 | / | / | 3.71±0.37 | 7.26±0.43 |
| Relative IF | 11.20±0.51 | 23.43±0.46 | 5.86±0.48 | 15.91±0.39 | 2.76±0.45 | 13.56±0.39 | 2.42±0.36 | 10.65±0.42 |
| Renorm. IF | 10.89±0.46 | 21.46±0.42 | 5.69±0.45 | 14.65±0.37 | 2.73±0.46 | 12.58±0.40 | 2.10±0.34 | 9.34±0.43 |
| TRAK | 11.42±0.49 | 23.59±0.46 | 5.78±0.48 | 15.87±0.39 | 2.93±0.46 | 13.62±0.38 | 2.20±0.38 | 10.33±0.42 |
| D-TRAK | 26.79±0.33 | 33.74±0.37 | 18.82±0.43 | 25.67±0.40 | 14.69±0.46 | 20.56±0.42 | 11.05±0.43 | 16.11±0.36 |
| **DAS** | **33.90±0.69** | **43.08±0.37** | **20.88±0.27** | **30.68±0.76** | **24.74±0.41** | **33.23±0.35** | **15.24±0.51** | **23.69±0.47** |

Table 2: LDS (%) on ArtBench-2/ArtBench-5 with timesteps (10 or 100)

| Method | ArtBench2 | | | | ArtBench5 | | | |
|---|---|---|---|---|---|---|---|---|
| | Validation | | Generation | | Validation | | Generation | |
| | 10 | 100 | 10 | 100 | 10 | 100 | 10 | 100 |
| Raw pixel (dot prod.) | 2.44±0.56 | | 2.60±0.84 | | 1.84±0.42 | | 2.77±0.80 | |
| Raw pixel (cosine) | 2.58±0.56 | | 2.71±0.86 | | 1.97±0.41 | | 3.22±0.78 | |
| CLIP similarity (dot prod.) | 7.18±0.70 | | 5.33±1.45 | | 5.29±0.45 | | 4.47±1.09 | |
| CLIP similarity (cosine) | 8.62±0.70 | | 8.66±1.31 | | 6.57±0.44 | | 6.63±1.14 | |
| Gradient (dot prod.) | 7.68±0.43 | 16.00±0.51 | 4.07±1.07 | 10.23±1.08 | 4.77±0.36 | 10.02±0.45 | 3.89±0.88 | 8.17±1.02 |
| Gradient (cosine) | 7.72±0.42 | 16.04±0.49 | 4.50±0.97 | 10.71±1.07 | 4.96±0.35 | 9.85±0.44 | 4.14±0.86 | 8.18±1.01 |
| TracInCP | 9.69±0.49 | 17.83±0.58 | 6.36±0.93 | 13.85±1.01 | 5.33±0.37 | 10.87±0.47 | 4.34±0.84 | 9.02±1.04 |
| GAS | 9.65±0.46 | 18.04±0.62 | 6.74±0.82 | 14.27±0.97 | 5.52±0.38 | 10.71±0.48 | 4.48±0.83 | 9.13±1.01 |
| Journey TRAK | / | / | 5.96±0.97 | 11.41±1.02 | / | / | 7.59±0.78 | 13.31±0.68 |
| Relative IF | 12.22±0.43 | 27.25±0.34 | 7.62±0.57 | 19.78±0.69 | 9.77±0.34 | 20.97±0.41 | 8.89±0.59 | 19.56±0.62 |
| Renorm. IF | 11.90±0.43 | 26.49±0.34 | 7.83±0.64 | 19.86±0.71 | 9.57±0.32 | 20.72±0.40 | 8.97±0.58 | 19.38±0.66 |
| TRAK | 12.26±0.42 | 27.28±0.34 | 7.78±0.59 | 20.02±0.69 | 9.79±0.33 | 21.03±0.42 | 8.79±0.59 | 19.54±0.61 |
| D-TRAK | 27.61±0.49 | 32.38±0.41 | 24.16±0.67 | 26.53±0.64 | 22.84±0.37 | 27.46±0.37 | 21.56±0.71 | 23.85±0.71 |
| **DAS** | **37.96±0.64** | **40.77±0.47** | **30.81±0.31** | **32.31±0.42** | **35.33±0.49** | **37.67±0.68** | **31.74±0.75** | **32.77±0.53** |

predicted model outputs across these subsets:

$$\text{LDS}(\tau, \boldsymbol{z}^{\text{test}}) := \rho\left(\{f(\boldsymbol{z}^{\text{test}}, \theta_m) : m \in [M]\}, \{g_\tau(\boldsymbol{z}^{\text{test}}, \mathbb{S}^m; \mathcal{D}) : m \in [M]\}\right) \tag{20}$$

where $\rho$ denotes the Spearman correlation and $\theta_m$ is the model trained on the $m$-th subset $\mathbb{S}^m$. In our evaluation, we adopt the output function setup from D-TRAK (Zheng et al., 2024), setting $f(\boldsymbol{z}^{\text{test}}, \theta)$ as the Simple Loss described in Eq. 3 for fairness. Although the output functions differ between D-TRAK and DAS, $f_{\text{DAS}}$ can also evaluate the Simple Loss change, where we elaborate on the rationale in Appendix D. Further details about LDS benchmarks are described in Appendix E.2.

## 5.3 Evaluation for Speed Up Techniques

The diffusion models used in our experiments are significantly complex, with parameter counts of 35.7M, 118.8M, and 25.5M respectively. These large dimensions pose considerable challenges in calculating the attribution score efficiently. To address this, we evaluate speed-up techniques as discussed in Section 4.3 on CIFAR-2. The results of these evaluations are reported in the Appendix E.4.

**Normalization.** We evaluate the normalization of gradients and residuals, as proposed in Eq. 17, to stabilize gradient variability across timesteps and enhance computational accuracy. By normalizing across generation before averaging, the performance for both DAS and D-TRAK improve (Table 4).

**Number of timesteps.** Computing DAS requires balancing effectiveness and computational efficiency, as more timesteps selected improve performance but increase costs. Experiment shows

that while increasing timesteps enhances LDS results (Table 5), using 100 or 10 timesteps achieves comparable performance to 1000 timesteps with much lower computational demands.

**Projection.** We apply the projection technique to reduce gradient dimensions. As Johnson & Lindenstrauss (1984) discussed, higher projection dimensions better preserve inner products but increase computational costs. Figure 2 shows that LDS scores for both D-TRAK and DAS improve with increasing $k$ before plateauing. Based on these results, we set $k = 32768$ as a default setting.

**Compress Model Parameters.** We explore techniques to reduce the gradient dimension by compressing. We conduct experiments by computing gradients on the up-block of U-Net, and the results in Table 6 show that this approach achieves competitive performance compared to the full model.

**Candidate Training Sample.** Another technique to speed up the calculation involves reducing the number of training samples considered. We use CLIP to select the 1,000 training samples most similar to the target, compute attribution scores only for this candidate set (assigning 0 to others), and calculate the LDS, with results in Table 7 validating its effectiveness.

## 5.4 Evaluating Output Function Effectiveness

We define the output function as $f_{\text{DAS}} = \epsilon_\theta(x_t^{\text{gen}}, t)$ to assess distribution shifts after data intervention. We argue that using Simple Loss introduces error as it involves the effect of data distribution during attribution. To validate this, we conduct a toy experiment using an unconditional DDPM trained on CIFAR-2 to generate 60 images. Each image is regenerated after removing 1,000 random training samples, producing 60 image pairs for which we compute $L^2$ distances. Denoising starts from timestep $T$, and we track average differences in loss and noise predictor outputs. Pearson correlation between $L^2$ distances and loss differences is 0.257, whereas for noise predictor differences, it reaches 0.485, indicating stronger alignment with image variations. Details are in Appendix H. This result aligns with intuition: if a model trained on cats and dogs generates a cat, removing all cat samples leads to a dog image under the same seed. Despite the drastic change, loss values may remain similar, as they reflect model optimization rather than generated content. Thus, loss differences fail to capture image changes, whereas noise predictor outputs effectively trace variations in diffusion model outputs.

## 5.5 Evaluating LDS for Various Attribution Methods

In this section, we evaluate the performance of the Diffusion Attribution Score (DAS) against existing attribution baselines applicable to our experimental settings, as outlined by Zheng et al. (2024), with detailed explanations provided in Appendix E.3. Our primary focus is on comparing with post-hoc data attribution methods. To ensure fair comparisons, we limit the use of acceleration techniques for DAS to projection only, excluding others like normalization. The results on CIFAR, ArtBench and CelebA, presented in Tables 1, 2, demonstrate that DAS consistently outperforms existing methods, where the result of CelebA is reported in the appendix in Appendix E.4 at Table 3.

Compared to D-TRAK, DAS shows substantial improvements. On a validation set utilizing 100 timesteps, DAS achieves improvements of +9.33% on CIFAR-2, +8.39% on ArtBench-2, and +5.1% on CelebA. In the generation set, the gains continue with +5.01% on CIFAR-2, +5.78% on ArtBench-2, and +9.21% on CelebA. Notably, DAS also achieves significant improvements on larger datasets like CIFAR10 and ArtBench5, outperforming D-TRAK by +12.67% and +10.21% on validation sets and +7.58% and +8.92% on generation samples.

Other methods generally underperform on larger datasets such as ArtBench5 and CIFAR10 compared to smaller datasets like CIFAR2 and ArtBench2. Conversely, our method performs better on ArtBench5 than on ArtBench2. Remarkably, our findings suggest that while with more timesteps for calculating gradients generally leads to a better approximation of the expectation $\mathbb{E}_t$, DAS, employing only a 10-timestep computation budget, still outperforms D-TRAK, which uses a 100-timestep budget in most cases, which underscores the effectiveness of our approach. Additionally, the modest improvements on CIFAR10 and CelebA may be attributed to the LDS setup for these datasets, which employs only one random seed per subset for training a model, whereas other datasets utilize three random seeds, potentially leading to inaccuracies in LDS evaluation. Another observation is that DAS performs better on the validation set than on the generation set. This could indicate

| Random | D-TRAK | DAS | | Random | D-TRAK | DAS |

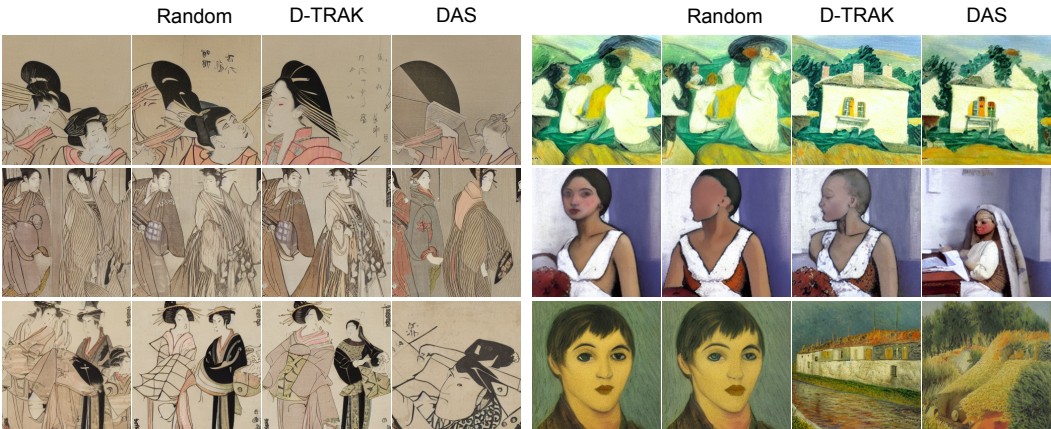

Figure 1: We conduct an visualization experiment to explore DAS effectiveness described in Sec 5.6. Removing the influential samples identified by DAS produces the most significant differences in the generated images after retraining the model. DAS is the most effective methods for attribution.

that the quality of generated images may have a significant impact on data attribution performance. However, further investigation is needed to validate this hypothesis.

## 5.6 Counter Factual Visualization Evaluation

To more intuitively assess the faithfulness of DAS, we conduct a counter factual visualization experiment. We use different attribution methods, including TRAK, D-TRAK and DAS, to identify the the top-1000 positive influencers on 60 generated images on ArtBench2 and CIFAR2. We sample 100 timesteps and a projection dimension of $k = 32768$ to identify the top-1000 influencers. These training samples identified by each method are subsequently removed and the model is retrained. Additionally, we conduct a baseline setting where 1,000 training images are randomly removed before retraining. We re-generate the images with same random seed and compare the $L^2$-Distance and CLIP cosine similarity between the origin and counter-factual images. For the pixel-wise $L^2$-Distance, D-TRAK yields values of 8.97 and 187.61 for CIFAR-2 and ArtBench-2, respectively, compared to TRAK's values of 5.90 and 168.37, while DAS results in values of 10.58 and 203.76. In terms of CLIP similarity, DAS achieves median similarities of 0.83 and 0.71 for ArtBench-2 and CIFAR-2, respectively, which are notably lower than TRAK's values of 0.94 and 0.84, as well as D-TRAK's values of 0.88 and 0.77, demonstrating the effectiveness of our method. An illustration of the experiment is in Figure 1, where removing the influencers detected by DAS results in a biggest difference compared to baseline methods. A detailed result in box-plot is reported in Appendix G.

## 6 Conclusion

In this paper, we introduce the Diffusion Attribution Score (DAS) to address the existing gap in data attribution methodologies for generative models. We conducted a comprehensive theoretical analysis to elucidate the inherent challenges in applying TRAK to diffusion models. Subsequently, we derived DAS theoretically based on the properties of diffusion models for attributing data throughout the entire generation process. We also discuss strategies to accelerate computations to extend DAS to large-scale diffusion models. Our extensive experimental evaluations on datasets such as CIFAR, CelebA, and ArtBench demonstrate that DAS consistently surpasses existing baselines in terms of Linear Datamodeling Score evaluation. This paper underscores the crucial role of data attribution in ensuring transparency in the use of diffusion models, especially when dealing with copyrighted or sensitive content. Looking forward, our future work aims to extend DAS to other generative models and real-world applications to further ascertain its effectiveness and applicability.

## 7 Acknowledgment

This work was supported in part by the Start-up Grant (No. 9610680) of the City University of Hong Kong, Young Scientist Fund (No. 62406265) of NSFC, and the Australian Research Council under Projects DP240101848 and FT230100549.

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

## A   PROOF OF EQUATION 13

In this section, we provide a detailed proof of Eq. 13. We utilize the Newton Method (Pregibon, 1981) to update the parameters in the diffusion model, where the parameter update $\theta'$ is defined as:

$$\theta' \leftarrow \theta + \boldsymbol{H}_{\theta_t}^{-1}(\mathcal{L}_{\text{Simple}}(\theta))\nabla_\theta \mathcal{L}_{\text{Simple}}(\theta), \tag{21}$$

Here, $\boldsymbol{H}_{\theta_t}(\mathcal{L}_{\text{Simple}}(\theta))$ represents the Hessian matrix, and $\nabla_\theta \mathcal{L}_{\text{Simple}}$ is the gradient w.r.t the Simple Loss in the diffusion model. At convergence, the model reaches the global optimum parameter estimate $\theta^*$, satisfying:

$$\boldsymbol{H}_{\theta_t}^{-1}(\mathcal{L}_{\text{Simple}}(\theta^*))\nabla_\theta \mathcal{L}_{\text{Simple}}(\theta^*) = 0. \tag{22}$$

Additionally, the Hessian matrix and gradient associated with the objective function at timestep $t$ are defined as:

$$\mathbf{H}_{\theta^*} = \nabla_\theta \boldsymbol{\epsilon}_{\theta^*}(\mathbb{S}_t, t)^\top \nabla_\theta \boldsymbol{\epsilon}_{\theta^*}(\mathbb{S}_t, t), \ \nabla_\theta \mathcal{L}(\theta^*) = \nabla_\theta \boldsymbol{\epsilon}_{\theta^*}(\mathbb{S}_t, t)^\top \boldsymbol{R}_t, \tag{23}$$

where $\boldsymbol{\epsilon}_{\theta^*}(\mathbb{S}_t, t) := [\boldsymbol{\epsilon}_{\theta^*}(\boldsymbol{x}_t^{(1)}, t), ..., \boldsymbol{\epsilon}_{\theta^*}(\boldsymbol{x}_t^{(n)}, t)]$ denotes a stacked output matrix on $\mathbb{S}$ at timestep $t$ and $\boldsymbol{R}_t := \text{diag}[\boldsymbol{\epsilon}_\theta(\boldsymbol{x}_t^{(i)}, t) - \boldsymbol{\epsilon}]$ is a diagonal matrix on $\mathbb{S}$ describing the residual among $\mathbb{S}$. Thus, the update defined in Eq. 21 around the optimum parameter $\theta^*$ is:

$$\theta' - \theta \leftarrow [\nabla_\theta \boldsymbol{\epsilon}_{\theta^*}(\mathbb{S}_t, t)^\top \nabla_\theta \boldsymbol{\epsilon}_{\theta^*}(\mathbb{S}_t, t)]^{-1}\nabla_\theta \boldsymbol{\epsilon}_{\theta^*}(\mathbb{S}_t, t)^\top \boldsymbol{R}_t. \tag{24}$$

Upon deleting a training sample $\boldsymbol{x}^{(i)}$ from $\mathbb{S}$, the counterfactual parameters $\theta_{\backslash i}^*$ can be estimated by applying a single step of Newton's method from the optimal parameter $\theta^*$ with the modified set $\mathbb{S}_{\backslash i}$, as follows:

$$\theta^* - \theta_{\backslash i}^* \leftarrow -[\nabla_\theta \boldsymbol{\epsilon}_{\theta^*}(\mathbb{S}_{\backslash i_t}, t)^\top \nabla_\theta \boldsymbol{\epsilon}_{\theta^*}(\mathbb{S}_{\backslash i_t}, t)]^{-1}\nabla_\theta \boldsymbol{\epsilon}_{\theta^*}(\mathbb{S}_{\backslash i_t}, t)^\top \boldsymbol{R}_{\backslash i_t}. \tag{25}$$

## B   PROOF OF EQUATION 15

In this section, we provide a detailed proof of Eq. 15. The Sherman-Morrison formula is defined as:

$$(\boldsymbol{A} + \boldsymbol{u}\boldsymbol{v}^\top)^{-1} = \boldsymbol{A}^{-1} - \frac{\boldsymbol{A}^{-1}\boldsymbol{u}\boldsymbol{v}^\top \boldsymbol{A}^{-1}}{1 + \boldsymbol{v}^\top \boldsymbol{A}^{-1}\boldsymbol{u}}. \tag{26}$$

Let $\boldsymbol{H} = \boldsymbol{G}_t(\mathbb{S})^\top \boldsymbol{G}_t(\mathbb{S})$ and $\boldsymbol{u} = \boldsymbol{g}_t(\boldsymbol{x}^{(i)})$. Applying Eq. 26 in Eq. 14, we derive:

$$[\boldsymbol{G}_t(\mathbb{S}_{\backslash i})^\top \boldsymbol{G}_t(\mathbb{S}_{\backslash i})]^{-1} = [\boldsymbol{H} - \boldsymbol{u}\boldsymbol{u}^\top]^{-1} = \boldsymbol{H}^{-1} + \frac{\boldsymbol{H}^{-1}\boldsymbol{u}\boldsymbol{u}^\top \boldsymbol{H}^{-1}}{1 - \boldsymbol{u}^\top \boldsymbol{H}^{-1}\boldsymbol{u}}. \tag{27}$$

Additionally, we have:

$$\boldsymbol{G}_t(\mathbb{S}_{\backslash i})^\top \boldsymbol{R}_{\backslash i_t} = \boldsymbol{G}_t(\mathbb{S})^\top \boldsymbol{R}_t - \boldsymbol{g}_t(\boldsymbol{x}^{(i)})^\top \boldsymbol{r}^{(i)}{}_t = -\boldsymbol{u}^\top \boldsymbol{r}_t^{(i)}. \tag{28}$$

Applying Eq. 27 and Eq. 28 to Eq. 25, we obtain:

$$\theta^* - \theta_{\backslash i}^* = [\boldsymbol{H}^{-1} + \frac{\boldsymbol{H}^{-1}\boldsymbol{u}\boldsymbol{u}^\top \boldsymbol{H}^{-1}}{1 - \boldsymbol{u}^\top \boldsymbol{H}^{-1}\boldsymbol{u}}]\boldsymbol{u}^\top \boldsymbol{r}_t^{(i)}. \tag{29}$$

Let $\alpha = \boldsymbol{u}^\top \boldsymbol{H}^{-1}\boldsymbol{u}$. Eq. 29 simplifies to:

$$\begin{aligned}
\theta^* - \theta_{\backslash i}^* &= \boldsymbol{H}^{-1}\boldsymbol{u} \cdot (1 + \frac{\alpha}{1-\alpha})\boldsymbol{r}_t^{(i)} \\
&= \boldsymbol{H}^{-1}\boldsymbol{u} \cdot \frac{1}{1-\alpha}\boldsymbol{r}_t^{(i)} \\
&= \frac{[\boldsymbol{G}_t(\mathbb{S})^\top \boldsymbol{G}_t(\mathbb{S})]^{-1}\boldsymbol{g}_t(\boldsymbol{x}^{(i)})\boldsymbol{r}_t^{(i)}}{1 - \boldsymbol{g}_t(\boldsymbol{x}^{(i)})^\top[(\boldsymbol{G}_t(\mathbb{S})^\top \boldsymbol{G}_t(\mathbb{S}))^{-1}\boldsymbol{g}_t(\boldsymbol{x}^{(i)})}.
\end{aligned} \tag{30}$$

## C    ALGORITHM OF DAS

In this section, we provide a algorithm about DAS in Algorithm 1.

---

**Algorithm 1** Diffusion Attribution Score

---

1:  **Input:** Learning algorithm $\mathcal{A}$, Training dataset $\mathbb{S}$ of size $n$, Training data dimension $p$, Maximum timesteps for diffusion model $T$, Unet output in diffusion model $\epsilon_\theta(\boldsymbol{x}, t)$, Projection dimension $k$, A standard gaussian noise $\epsilon \sim \mathcal{N}(0, \boldsymbol{I})$, Normalization and average method $N$, A generated sample $\boldsymbol{z}^{\text{gen}}$
2:  **Output:** Matrix of attribution scores $\boldsymbol{T} \in \boldsymbol{R}^n$
3:  $\theta^* \leftarrow \mathcal{A}(\mathbb{S})$               ▷ Train a diffusion model on $\mathbb{S}$
4:  $\boldsymbol{P} \sim \mathcal{N}(0,1)^{p \times k}$             ▷ Sample projection matrix
5:  $\boldsymbol{R} \leftarrow 0_{n \times n}$
6:  **for** $i = 1$ to $n$ **do**
7:   **for** $t = 1$ to $T$ **do**
8:    $\epsilon \sim \mathcal{N}(0,1)^p$          ▷ Sample a gaussian noise
9:    $\boldsymbol{g}_t(\boldsymbol{x}^{(i)}) \leftarrow \boldsymbol{P}^\top \nabla_\theta \epsilon_{\theta^*}(\boldsymbol{x}_t^{(i)}, t)$    ▷ Compute gradient on training set and project
10:   $\boldsymbol{r}_t^{(i)} \leftarrow \epsilon_{\theta^*}(\boldsymbol{x}_t^{(i)}, t) - \epsilon$       ▷ Compute residual term
11:  **end for**
12:  $\overline{\boldsymbol{g}}(\boldsymbol{x}^{(i)}) = N(\boldsymbol{g}_t(\boldsymbol{x}^{(i)}))$      ▷ Normalize projected gradient term
13:  $\overline{\boldsymbol{r}}^{(i)} = N(\boldsymbol{r}_t^{(i)})$         ▷ Normalize residual term
14: **end for**
15: $\overline{\boldsymbol{G}}(\mathbb{S}) \leftarrow [\overline{\boldsymbol{g}}(\boldsymbol{x}^{(i)}), ..., \overline{\boldsymbol{g}}(\boldsymbol{x}^{(n)})]^\top$
16: $\overline{\boldsymbol{R}} \leftarrow \text{diag}(\overline{\boldsymbol{r}}^{(1)}, ..., \overline{\boldsymbol{r}}^{(n)})$
17: **for** $t = 1$ to $T$ **do**
18:  $\epsilon \sim \mathcal{N}(0,1)^p$          ▷ Sample a gaussian noise
19:  $\boldsymbol{g}_t(\boldsymbol{x}^{\text{gen}}) \leftarrow \boldsymbol{P}^\top \nabla_\theta \epsilon_{\theta^*}(\boldsymbol{x}_t^{\text{gen}}, t)$   ▷ Compute gradient for generated sample and project
20: **end for**
21: $\overline{\boldsymbol{g}}(x^{\text{gen}}) = N(\boldsymbol{g}_t(x^{\text{gen}}))$
22: $\boldsymbol{T} \leftarrow ||\frac{\overline{\boldsymbol{g}}(\boldsymbol{x}^{\text{gen}})^\top (\overline{\boldsymbol{G}}(\mathbb{S})^\top \overline{\boldsymbol{G}}(\mathbb{S}))^{-1} \overline{\boldsymbol{G}}(\mathbb{S}) \overline{\boldsymbol{R}}}{1 - \overline{\boldsymbol{G}}(\mathbb{S})^\top (\overline{\boldsymbol{G}}(\mathbb{S})^\top \overline{\boldsymbol{G}}(\mathbb{S}))^{-1} \overline{\boldsymbol{G}}(\mathbb{S})}||^2$    ▷ Compute attribution matrix
23: **return** $(T)$

---

## D    EXPLANATION ABOUT THE CHOICE OF MODEL OUTPUT FUNCTION IN LDS EVALUATION

In the LDS framework, we evaluate the ranking correlation between the ground-truth and the predicted model outputs following a data intervention. In our evaluations, the output function used in JourneyTRAK and D-TRAK is shown to provide the same ranking as our DAS output function. Below, we provide a detailed explanation of this alignment.

Defining the function $f(\boldsymbol{z}, \theta)$ as $\mathcal{L}_{\text{Simple}}$, D-TRAK predicts the change in output as:

$$f(\boldsymbol{z}^{\text{gen}}, \theta) - f(\boldsymbol{z}^{\text{gen}}, \theta_m) = \mathbb{E}_{\epsilon, t}[||\epsilon - \epsilon_\theta(\boldsymbol{z}_t^{\text{gen}}, t)||^2] - \mathbb{E}_\epsilon[||\epsilon - \epsilon_{\theta_m}(\boldsymbol{z}_t^{\text{gen}}, t)||^2], \quad (31)$$

where $\theta$ and $\theta_m$ represent the models trained on the full dataset $\mathbb{S}$ and a subset $\mathbb{S}_m$, respectively. With the retraining of the model on subset $\mathbb{S}_m$ while fixing all randomness, the real change can be computed as:

$$f(\boldsymbol{z}^{\text{gen}}, \theta) - f(\boldsymbol{z}^{\text{gen}}, \theta_m) = \mathbb{E}_{\epsilon, t}[||\epsilon_\theta(\boldsymbol{z}_t^{\text{gen}}, t) - \epsilon_{\theta_m}(\boldsymbol{z}_t^{\text{gen}}, t)||^2]$$
$$+ 2\mathbb{E}_\epsilon[(\epsilon_\theta(\boldsymbol{z}_t^{\text{gen}}, t) - \epsilon_{\theta_m}(\boldsymbol{z}_t^{\text{gen}}, t)) \cdot (\epsilon - \epsilon_{\theta_m}(\boldsymbol{z}_t^{\text{gen}}, t))]. \quad (32)$$

The first expectation represents the predicted output change in DAS as well as the ground truth output, while the term $(\epsilon - \epsilon_{\theta_m}(\boldsymbol{z}_t^{\text{gen}}, t))$ is recognized as the error of the MMSE estimator.

In our evaluations, the sampling noise $\epsilon$ is fixed and $\epsilon_\theta(\boldsymbol{z}_t^{\text{gen}}, t)$ is a given value since the model $\theta$ is frozen. Therefore, the second term equals zero, following the orthogonality principle. It can be stated as a more general result that,

$$\forall f, \mathbb{E}[f(\epsilon_{\theta_m}, \boldsymbol{z}^{\text{gen}}) \cdot (\epsilon - \epsilon_{\theta_m}(\boldsymbol{z}^{\text{gen}}))] = 0. \quad (33)$$

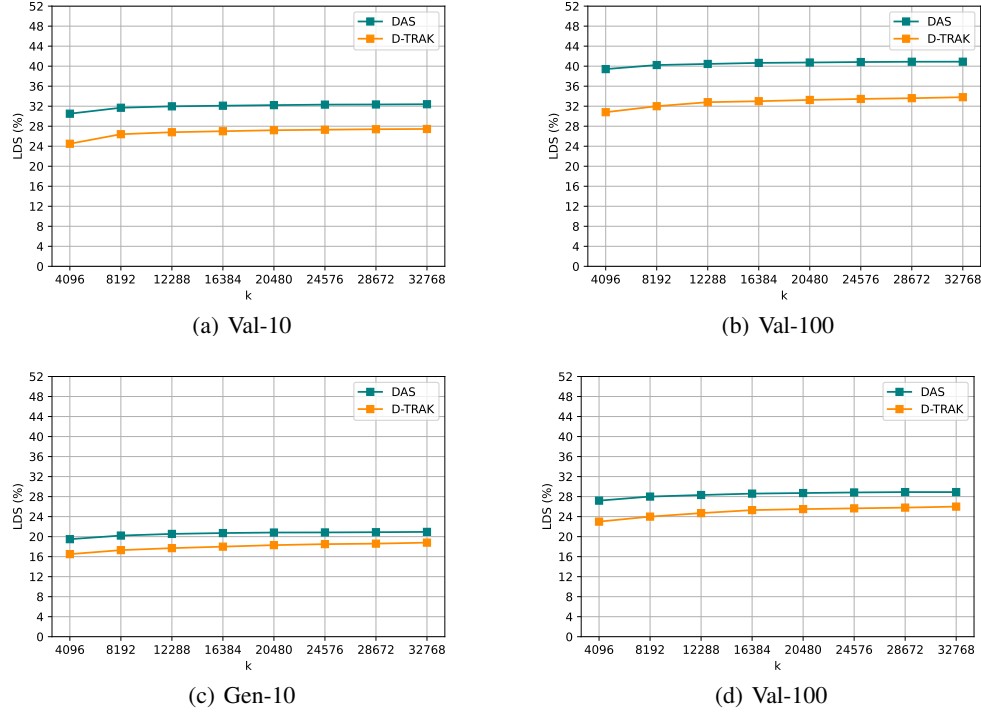

Figure 2: The LDS(%) on CIFAR-2 under different projection dimension $k$. We consider 10 and 100 timesteps selected to be evenly spaced within the interval $[1, T]$, which are used to approximate the expectation $\mathbb{E}_t$. For each sampled timestep, we sample one standard Gaussian noise $\epsilon \sim \mathcal{N}(\epsilon|0, I)$ to approximate the expectation $\mathbb{E}_\epsilon$.

The error $(\epsilon - \epsilon_{\theta_m}(z^{\text{gen}}))$ must be orthogonal to any estimator $f$. If not, we could use $f$ to construct an estimator with a lower MSE than $\epsilon_{\theta_m}(x_{gen})$, contradicting our assumption that $(\epsilon - \epsilon_{\theta_m}(z^{\text{gen}}))$ is the MMSE estimator. Applications of the orthogonality principle in diffusion models have been similarly proposed (Kong et al., 2024; Banerjee et al., 2005). Thus, given that we have a frozen model $\theta$, fixed sampling noise $\epsilon$ and a specific generated sample $z^{\text{gen}}$, the change in the output function in DAS provides the same rank as the one in JourneyTRAK and D-TRAK.

# E IMPLEMENTATION DETAILS

## E.1 DATASETS AND MODELS

**CIFAR10(32×32)**. The CIFAR-10 dataset, introduced by Krizhevsky (2009), consists of 50,000 training images across various classes. For the Linear Datamodeling Score evaluation, we utilize a subset of 1,000 images randomly selected from CIFAR-10's test set. To manage computational demands effectively, we also create a smaller subset, CIFAR-2, which includes 5,000 training images and 1,000 validation images specifically drawn from the "automobile" and "horse" categories of CIFAR-10's training and test sets, respectively.

In our CIFAR experiments, we employ the architecture and settings of the Denoising Diffusion Probabilistic Models (DDPMs) as outlined by Ho et al. (2020). The model is configured with approximately 35.7 million parameters ($d = 35.7 \times 10^6$ for $\theta \in \mathbb{R}^d$). We set the maximum number of timesteps ($T$) at 1,000 with a linear variance schedule for the forward diffusion process, beginning at $\beta_1 = 10^{-4}$ and escalating to $\beta_T = 0.02$. Additional model specifications include a dropout rate of 0.1 and the use of the AdamW optimizer (Loshchilov & Hutter, 2019) with a weight decay of $10^{-6}$. Data augmentation techniques such as random horizontal flips are employed to enhance model robustness. The training process spans 200 epochs with a batch size of 128, using a cosine annealing

Figure 3: The LDS(%) on CIFAR-2 varies across different checkpoints. We analyze the data using 10 and 100 timesteps, evenly spaced within the interval $[1, T]$, to approximate the expectation $\mathbb{E}_t$. At each sampled timestep, we introduce one standard Gaussian noise $\epsilon \sim \mathcal{N}(\mathbf{0}, \mathbf{I})$ to approximate the expectation $\mathbb{E}_\epsilon$. We set the projection dimension $k = 32768$.

learning rate schedule that incorporates a warm-up period covering 10% of the training duration, beginning from an initial learning rate of $10^{-4}$. During inference, new images are generated using the 50-step Denoising Diffusion Implicit Models (DDIM) solver (Song et al., 2021a).

**CelebA(64×64)**. We selected 5,000 training samples and 1,000 validation samples from the original training and test sets of CelebA (Liu et al., 2015), respectively. Following the preprocessing method described by Song et al. (2021b), we first center-cropped the images to 140×140 pixels and then resized them to 64×64 pixels. For the CelebA experiments, we adapted the architecture to accommodate a 64×64 resolution while employing a similar unconditional DDPM implementation as used for CIFAR-10. However, the U-Net architecture was expanded to 118.8 million parameters to better capture the increased complexity of the CelebA dataset. The hyperparameter, including the variance schedule, optimizer settings, and training protocol, were kept consistent with those used for the CIFAR-10 experiments.

**ArtBench(256×256)**. ArtBench (Liao et al., 2022) is a dataset specifically designed for generating artwork, comprising 60,000 images across 10 unique artistic styles. Each style contributes 5,000 training images and 1,000 testing images. We introduce two subsets from this dataset for focused evaluation: ArtBench-2 and ArtBench-5. ArtBench-2 features 5,000 training and 1,000 validation images selected from the "post-impressionism" and "ukiyo-e" styles, extracted from a total of 10,000 training and 2,000 test images. ArtBench-5 includes 12,500 training and 1,000 validation images drawn from a larger pool of 25,000 training and 5,000 test images across five styles: "post-impressionism," "ukiyo-e," "romanticism," "renaissance," and "baroque."

For our experiments on ArtBench, we fine-tune a Stable Diffusion model (Rombach et al., 2022) using Low-Rank Adaptation (LoRA) (Hu et al., 2022) with a rank of 128, amounting to 25.5 million parameters. We adapt a pre-trained Stable Diffusion checkpoint from a resolution of 512×512 to 256×256 to align with the ArtBench specifications. The model is trained conditionally using textual prompts specific to each style, such as "a class painting," e.g., "a romanticism painting." We set the dropout rate at 0.1 and employ the AdamW optimizer with a weight decay of $10^{-6}$. Data augmentation is performed via random horizontal flips. The training is conducted over 100 epochs with a batch size of 64, under a cosine annealing learning rate schedule that includes a 0.1 fraction warm-up period starting from an initial rate of $3 \times 10^{-4}$. During the inference phase, we generate new images using the 50-step DDIM solver with a classifier-free guidance scale of 7.5 (Ho & Salimans, 2021).

### E.2 LDS EVALUATION SETUP

Various methods are available for evaluating data attribution techniques, including the leave-one-out influence method (Koh & Liang, 2017; Basu et al., 2021) and Shapley values (Lundberg & Lee, 2017). In this paper, we use the Linear Datamodeling Score (LDS) (Ilyas et al., 2022) to assess data attribution methods. Given a model trained $\theta$ on dataset $\mathbb{S}$, LDS evaluates the effectiveness of a data attribution method $\tau$ by initially sampling a sub-dataset $\mathbb{S}' \subset \mathbb{S}$ and retraining a model $\theta'$ on $\mathbb{S}'$. The attribution-based output prediction for an interested sample $\boldsymbol{z}^{\text{test}}$ is then calculated as:

$$g_\tau(\boldsymbol{z}^{\text{test}}, \mathbb{S}', \mathbb{S}) := \sum_{\boldsymbol{z}^{(i)} \in \mathbb{S}'} \tau(\boldsymbol{z}^{\text{test}}, \mathbb{S})^{(i)} \tag{34}$$

The underlying premise of LDS is that the predicted output $g_\tau(\boldsymbol{z}, \mathbb{S}', \mathbb{S})$ should correspond closely to the actual model output $f(\boldsymbol{z}^{\text{test}}, \theta')$. To validate this, LDS samples $M$ subsets of fixed size and

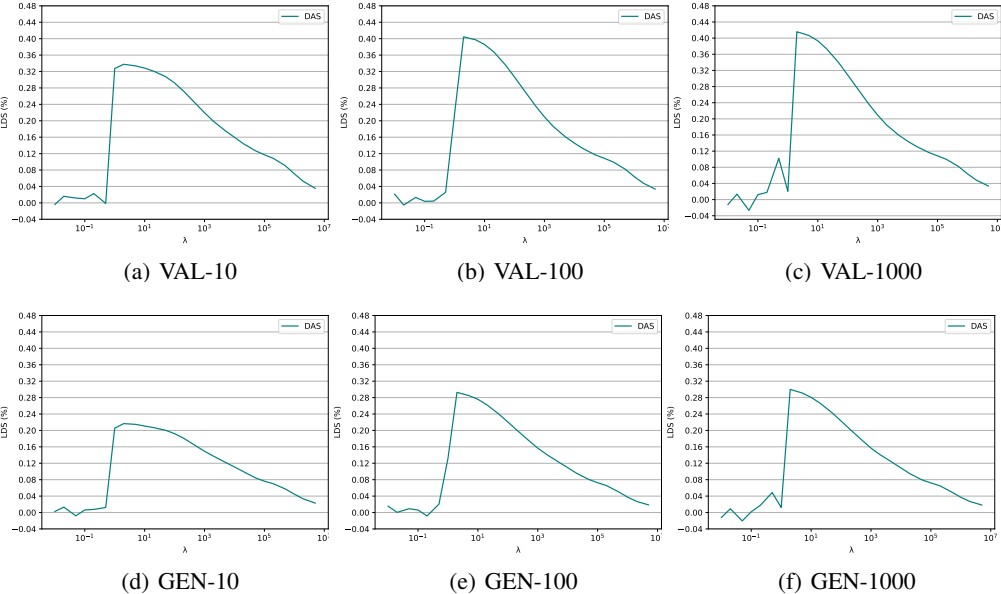

Figure 4: LDS (%) on CIFAR-2 under different $\lambda$. We consider 10, 100, and 1000 timesteps selected to be evenly spaced within the interval $[1, T]$, which are used to approximate the expectation $\mathbb{E}_t$. We set $k = 4096$.

predicted model outputs across these subsets:

$$\text{LDS}(\tau, \boldsymbol{z}^{\text{test}}) := \rho\left(\{f(\boldsymbol{z}^{\text{test}}, \theta_m) : m \in [M]\}, \{g_\tau(\boldsymbol{z}^{\text{test}}, \mathbb{S}^m; \mathcal{D}) : m \in [M]\}\right) \quad (35)$$

where $\rho$ denotes the Spearman correlation and $\theta_m$ is the model trained on the $m$-th subset $\mathbb{S}^m$. For the LDS evaluation, we construct 64 distinct subsets $\mathbb{S}_m$ from the training dataset $\mathbb{S}$, each constituting 50% of the total training set size. For CIFAR-2, ArtBench-2, and ArtBench-5, three models per subset are trained using different random seeds for robustness, while for CIFAR-10 and CelebA, a single model is trained per subset. A validation set, comprising 1,000 samples each from the original test set and a generated dataset, serves as $\boldsymbol{z}^{\text{test}}$ for LDS calculations. Specifically, we evaluate the Simple Loss $\mathcal{L}_{\text{Simple}}(\boldsymbol{z}, \theta)$ as defined in Eq 3 for samples of interest from both the validation and generation sets. To better approximate the expectation $\mathbb{E}_t$, we utilize 1,000 timesteps, evenly spaced within the range $[1, T]$. At each timestep, three instances of standard Gaussian noise $\boldsymbol{\epsilon} \sim \mathcal{N}(0, \boldsymbol{I})$ are introduced to approximate the expectation $\mathbb{E}_{\boldsymbol{\epsilon}}$. The calculated LDS values are then averaged across the selected samples from both the validation and generation sets to determine the overall LDS performance.

### E.3 BASELINES

In this paper, our focus is primarily on post-hoc data attribution, which entails applying attribution methods after the completion of model training. These methods are particularly advantageous as they do not impose additional constraints during the model training phase, making them well-suited for practical applications (Ribeiro et al., 2016).

Following the work of Hammoudeh & Lowd (2024), we evaluate various attribution baselines that are compatible with our experimental framework. We exclude certain methods that are not feasible for our settings, such as the Leave-One-Out approach (Cook, 1977) and the Shapley Value method (Shapley et al., 1953; Ghorbani & Zou, 2019). These methods, although foundational, do not align well with the requirements of DDPMs due to their intensive computational demands and model-specific limitations. Additionally, we do not consider techniques like Representer Point (Yeh et al., 2018), which are tailored for specific tasks and models, and thus are incompatible with DDPMs. Moreover, we disregard HYDRA (Chen et al., 2021), which, although related to Trac-

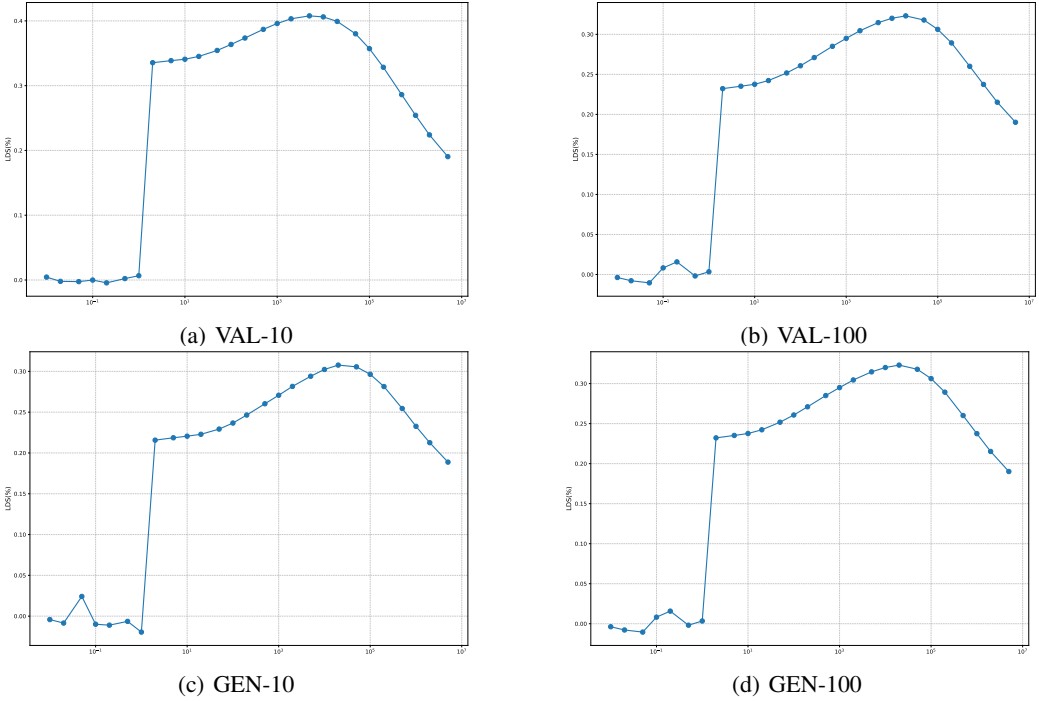

(a) VAL-10                                       (b) VAL-100

(c) GEN-10                                       (d) GEN-100

Figure 5: LDS (%) on ArtBench-2 under different $\lambda$. We consider 10 and 100 timesteps selected to be evenly spaced within the interval $[1, T]$, which are used to approximate the expectation $\mathbb{E}_t$. We set $k = 32768$.

InCP (Pruthi et al., 2020), compromises precision for incremental speed improvements as critiqued by Hammoudeh & Lowd (2024).

Two works focus on diffusion model that also fall outside our framework. Dai & Gifford (2023) propose a method for training data attribution on diffusion models using machine unlearning; however, their approach necessitates a specific machine unlearning training process, making it non-post-hoc and thus unsuitable for standard settings. Similarly, Wang et al. (2023) acknowledge the current challenges in conducting post-hoc training influence analysis with existing methods. They suggest an alternative termed "customization," which involves adapting or tuning a pretrained text-to-image model through a specially designed training procedure.

Building upon recent advancements, Park et al. (2023) introduced an innovative estimator that leverages a kernel matrix analogous to the Fisher Information Matrix (FIM), aiming to linearize the model's behavior. This approach integrates classical random projection techniques to expedite the computation of Hessian-based influence functions (Koh & Liang, 2017), which are typically computationally intensive. Zheng et al. (2024) adapted TRAK to diffusion models, empirically designing the model output function. Intriguingly, they reported that the theoretically designed model output function in TRAK performs poorly in unsupervised settings within diffusion models. However, they did not provide a theoretical explanation for these empirical findings, leaving a gap in understanding the underlying mechanics.

Our study concentrates on retraining-free methods, which we categorize into three distinct types: similarity-based, gradient-based (without kernel), and gradient-based (with kernel) methods. For similarity-based approaches, we consider Raw pixel similarity and CLIP similarity (Radford et al., 2021). The gradient-based methods without a kernel include techniques such as Gradient (Charpiat et al., 2019), TracInCP (Pruthi et al., 2020) and GAS (Hammoudeh & Lowd, 2022). In the domain of gradient-based methods with a kernel, we explore several methods including D-TRAK (Zheng et al., 2024), TRAK (Park et al., 2023), Relative Influence (Barshan et al., 2020), Renormalized Influence (Hammoudeh & Lowd, 2022), and Journey TRAK (Georgiev et al., 2023).

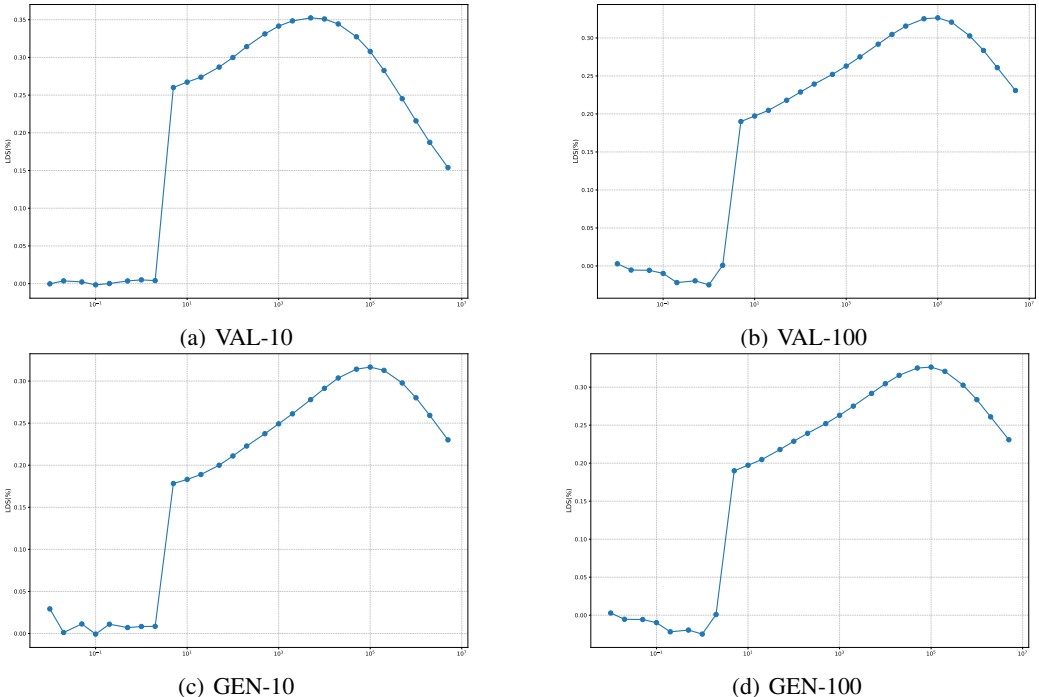

Figure 6: LDS (%) on ArtBench-5 under different $\lambda$. We consider 10 and 100 timesteps selected to be evenly spaced within the interval $[1, T]$, which are used to approximate the expectation $\mathbb{E}_t$. We set $k = 32768$.

We next provide definition and implementation details of the baselines used in Section 5.5.

**Raw pixel.** This method employs a naive similarity-based approach for data attribution by using the raw image data itself as the representation. Specifically, for experiments on ArtBench, which utilizes latent diffusion models (Rombach et al., 2022), we represent the images through the VAE encodings (Van Den Oord et al., 2017) of the raw image. The attribution score is calculated by computing either the dot product or cosine similarity between the sample of interest and each training sample, facilitating a straightforward assessment of similarity based on pixel values.

**CLIP Similarity.** This method represents another similarity-based approach to data attribution. Each sample is encoded into an embedding using the CLIP model (Radford et al., 2021), which captures semantic and contextual nuances of the visual content. The attribution score is then determined by computing either the dot product or cosine similarity between the CLIP embedding of the target sample and those of the training samples. This method leverages the rich representational power of CLIP embeddings to ascertain the contribution of training samples to the generation or classification of new samples.

**Gradient.** This method employs a gradient-based approach to estimate the influence of training samples, as described by Charpiat et al. (2019). The attribution score is calculated by taking the dot product or cosine similarity between the gradients of the sample of interest and those of each training sample. This technique quantifies how much the gradient (indicative of the training sample's influence on the loss) of a particular training sample aligns with the gradient of the sample of interest, providing insights into which training samples most significantly affect the model's output.

$$\tau(\boldsymbol{z}, \mathbb{S})^i = (\boldsymbol{P}^\top \nabla_\theta \mathcal{L}_{\text{Simple}}(\boldsymbol{x}, \theta))^\top \cdot (\boldsymbol{P}^\top \nabla_\theta \mathcal{L}_{\text{Simple}})(\boldsymbol{x}^{(i)}, \theta^*),$$

$$\tau(\boldsymbol{z}, \mathbb{S})^i = \frac{(\boldsymbol{P}^\top \nabla_\theta \mathcal{L}_{\text{Simple}}(\boldsymbol{x}, \theta))^\top \cdot (\boldsymbol{P}^\top \nabla_\theta \mathcal{L}_{\text{Simple}}(\boldsymbol{x}^{(i)}, \theta))}{||\boldsymbol{P}^\top \nabla_\theta \mathcal{L}_{\text{Simple}}(\boldsymbol{x}, \theta)||^\top ||\boldsymbol{P}^\top \nabla_\theta \mathcal{L}_{\text{Simple}}(\boldsymbol{x}^{(i)}, \theta)||}.$$

Table 3: LDS (%) on CelebA with timesteps (10 or 100)

| | Results on CelebA | | | |
|---|---|---|---|---|
| Method | Validation | | Generation | |
| | 10 | 100 | 10 | 100 |
| Raw pixel (dot prod.) | 5.58±0.73 | | -4.94±1.58 | |
| Raw pixel (cosine) | 6.16±0.75 | | -4.38±1.63 | |
| CLIP similarity (dot prod.) | 8.87±1.14 | | 2.51±1.13 | |
| CLIP similarity (cosine) | 10.92±0.87 | | 3.03±1.13 | |
| Gradient (dot prod.) | 3.82±0.50 | 4.89±0.65 | 3.83±1.06 | 4.53±0.84 |
| Gradient (cosine) | 3.65±0.52 | 4.79±0.68 | 3.86±0.96 | 4.40±0.86 |
| TracInCP | 5.14±0.75 | 4.89±0.86 | 5.18±1.05 | 4.50±0.93 |
| GAS | 5.44±0.68 | 5.19±0.64 | 4.69±0.97 | 3.98±0.97 |
| Journey TRAK | / | / | 6.53±1.06 | 10.87±0.84 |
| Relative IF | 11.10±0.51 | 19.89±0.50 | 6.80±0.77 | 14.66±0.70 |
| Renorm. IF | 11.01±0.50 | 18.67±0.51 | 6.74±0.82 | 13.24±0.71 |
| TRAK | 11.28±0.47 | 20.02±0.47 | 7.02±0.89 | 14.71±0.70 |
| D-TRAK | 22.83±0.51 | 28.69±0.44 | 16.84±0.54 | 21.47±0.48 |
| DAS | **29.38±0.51** | **33.79±0.23** | **28.73±0.49** | **30.68±0.31** |

**TracInCP.** We implement the TracInCP estimator, as outlined by Pruthi et al. (2020), which quantifies the influence of training samples using the following formula:

$$\tau(\boldsymbol{z}, \mathbb{S})^i = \frac{1}{C} \Sigma_{c=1}^{C} (\boldsymbol{P}_c^\top \nabla_\theta \mathcal{L}_{\text{Simple}}(\boldsymbol{x}, \theta^c))^\top \cdot (\boldsymbol{P}_c^\top \nabla_\theta \mathcal{L}_{\text{Simple}}(\boldsymbol{x}^i, \theta^c)),$$

where $C$ represents the number of model checkpoints selected evenly from the training trajectory, and $\theta^c$ denotes the model parameters at each checkpoint. For our analysis, we select four specific checkpoints along the training trajectory to ensure a comprehensive evaluation of the influence over different phases of learning. For example, in the CIFAR-2 experiment, the chosen checkpoints occur at epochs 50, 100, 150, and 200, capturing snapshots of the model's development and adaptation.

**GAS.** The GAS method is essentially a "renormalized" version of TracInCP that employs cosine similarity for estimating influence, rather than relying on raw dot products. This method was introduced by Hammoudeh & Lowd (2022) and aims to refine the estimation of influence by normalizing the gradients. This approach allows for a more nuanced comparison between gradients, considering not only their directions but also normalizing their magnitudes to focus solely on the directionality of influence.

**TRAK.** The retraining-free version of TRAK (Park et al., 2023) utilizes a model's trained state to estimate the influence of training samples without the need for retraining the model at each evaluation step. This version is implemented using the following equations:

$$\boldsymbol{\Phi}_{\text{TRAK}} = \left[\boldsymbol{\Phi}(\boldsymbol{x}^1), \cdots, \boldsymbol{\Phi}(\boldsymbol{x}^N)\right]^\top, \text{ where } \boldsymbol{\Phi}(\boldsymbol{x}) = \boldsymbol{P}^\top \nabla_\theta \mathcal{L}_{\text{Simple}}(\boldsymbol{x}, \theta),$$

$$\tau(\boldsymbol{z}, \mathbb{S})^i = (\boldsymbol{P}^\top \nabla_\theta \mathcal{L}_{\text{Simple}}(\boldsymbol{x}, \theta))^\top \cdot \left(\boldsymbol{\Phi}_{\text{TRAK}}^\top \boldsymbol{\Phi}_{\text{TRAK}} + \lambda \boldsymbol{I}\right)^{-1} \cdot \boldsymbol{P}^\top \nabla_\theta \mathcal{L}_{\text{Simple}}(\boldsymbol{x}^i, \theta),$$

where $\lambda \boldsymbol{I}$ is included for numerical stability and regularization. The impact of this term is further explored in Appendix F.

**D-TRAK.** Similar to TRAK, as elaborated in Section 3, we adapt the D-TRAK (Zheng et al., 2024) as detailed in Eq 6. We implement the model output function $f(\boldsymbol{z}, \theta)$ as $\mathcal{L}_{\text{Square}}$. The D-TRAK is implemented using the following equations:

$$\boldsymbol{\Phi}_{\text{D-TRAK}} = \left[\boldsymbol{\Phi}(\boldsymbol{x}^1), \cdots, \boldsymbol{\Phi}(\boldsymbol{x}^N)\right]^\top, \text{ where } \boldsymbol{\Phi}(\boldsymbol{x}) = \boldsymbol{P}^\top \nabla_\theta \mathcal{L}_{\text{Simple}}(\boldsymbol{x}, \theta),$$

$$\tau(\boldsymbol{z}, \mathbb{S})^i = (\boldsymbol{P}^\top \nabla_\theta \mathcal{L}_{\text{Simple}}(\boldsymbol{x}, \theta))^\top \cdot \left(\boldsymbol{\Phi}_{\text{TRAK}}^\top \boldsymbol{\Phi}_{\text{TRAK}} + \lambda \boldsymbol{I}\right)^{-1} \cdot \boldsymbol{P}^\top \nabla_\theta \mathcal{L}_{\text{Simple}}(\boldsymbol{x}^i, \theta),$$

where $\lambda I$ is also included for numerical stability and regularization as TRAK. Additionally, the output function $f(\boldsymbol{z}, \theta)$ could be replaced to other functions.

Table 4: We compare D-TRAK and our methods DAS with the normalization and without normalization on CIFAR2. Besides, we also select 10, 100 and 1000 timesteps evenly spaced within the interval $[1, T]$ and calculate the average of LDS(%) among the timesteps.

| Method | Normalization | Validation | | | Generation | | |
|--------|---------------|-----|-----|------|-----|-----|------|
| | | 10 | 100 | 1000 | 10 | 100 | 1000 |
| D-TRAK | No Normalization | 24.78 | 30.81 | 32.37 | 16.20 | 22.62 | 23.94 |
| | Normalization | 26.11 | 31.50 | 32.51 | 17.09 | 22.92 | 24.10 |
| DAS | No Normalization | 33.04 | 42.02 | 43.13 | 20.01 | 29.58 | 30.58 |
| | Normalization | 33.77 | 42.26 | 43.28 | 21.24 | 29.60 | 30.87 |

**Relative Influence.** Barshan et al. (2020) introduce the $\theta$-relative influence functions estimator, which normalizes the influence functions estimator from Koh & Liang (2017) by the magnitude of the Hessian-vector product (HVP). This normalization enhances the interpretability of influence scores by adjusting for the impact magnitude. We have adapted this method to our experimental framework by incorporating scalability optimizations from TRAK. The adapted equation for the Relative Influence is formulated as follows:

$$\tau(\boldsymbol{z}, \mathbb{S})^{(i)} = \frac{(\boldsymbol{P}^\top \nabla_\theta \mathcal{L}_{\text{Simple}}(\boldsymbol{x}, \theta))^\top \cdot \left(\boldsymbol{\Phi}_{\text{TRAK}}^\top \boldsymbol{\Phi}_{\text{TRAK}} + \lambda \boldsymbol{I}\right)^{-1} \cdot \boldsymbol{P}^\top \nabla_\theta \mathcal{L}_{\text{Simple}}(\boldsymbol{x}^{(i)}, \theta^*)}{|| \left(\boldsymbol{\Phi}_{\text{TRAK}}^\top \boldsymbol{\Phi}_{\text{TRAK}} + \lambda I\right)^{-1} \cdot \boldsymbol{P}^\top \nabla_\theta \mathcal{L}_{\text{Simple}}(\boldsymbol{x}^{(i)}, \theta^*)||}.$$

**Renormalized Influence.** Hammoudeh & Lowd (2022) propose a method to renormalize influence by considering the magnitude of the training sample's gradients. This approach emphasizes the relative strength of each sample's impact on the model, making the influence scores more interpretable and contextually relevant. We have adapted this method to our settings by incorporating TRAK's scalability optimizations, which are articulated as:

$$\tau(\boldsymbol{z}, \mathbb{S})^{(i)} = \frac{(\boldsymbol{P}^\top \nabla_\theta \mathcal{L}_{\text{Simple}}(\boldsymbol{x}, \theta))^\top \cdot \left(\boldsymbol{\Phi}_{\text{TRAK}}^\top \boldsymbol{\Phi}_{\text{TRAK}} + \lambda \boldsymbol{I}\right)^{-1} \cdot \boldsymbol{P}^\top \nabla_\theta \mathcal{L}_{\text{Simple}}(\boldsymbol{x}^{(i)}, \theta)}{||\boldsymbol{P}^\top \nabla_\theta \mathcal{L}_{\text{Simple}}(\boldsymbol{x}^{(i)}, \theta)||}.$$

**Journey TRAK.** Journey TRAK (Georgiev et al., 2023) focuses on attributing influence to noisy images $\boldsymbol{x}_t$ at a specific timestep $t$ throughout the generative process. In contrast, our approach aims to attribute the final generated image $\boldsymbol{x}^{\text{gen}}$, necessitating an adaptation of their method to our context. We average the attributions across the generation timesteps, detailed in the following equation:

$$\tau(\boldsymbol{z}, \mathbb{S})^{(i)} = \frac{1}{T'} \Sigma_{t=1}^{T'} (\boldsymbol{P}^\top \nabla_\theta \mathcal{L}_{\text{Simple}}^t(\boldsymbol{x}_t, \theta))^\top \cdot \left(\boldsymbol{\Phi}_{\text{TRAK}}^\top \boldsymbol{\Phi}_{\text{TRAK}} + \lambda \boldsymbol{I}\right)^{-1} \cdot \boldsymbol{P}^\top \nabla_\theta \mathcal{L}_{\text{Simple}}(\boldsymbol{x}^{(i)}, \theta),$$

where $T'$ represents the number of inference steps, set at 50, and $\boldsymbol{x}_t$ denotes the noisy image generated along the sampling trajectory.

### E.4 EXPERIMENTS RESULT

In this subsection, we present the outcomes of experiments detailed in Section 5.3. The results, which illustrate the effectiveness of various techniques designed to expedite computational processes in data attribution, are summarized in several tables and figures. These include Table 4, Table 5, Table 6, and Table 7, as well as Figure 2. Each of these displays key findings relevant to the specific speed-up technique tested, providing a comprehensive view of their impacts on attribution performance.

## F ABLATION STUDIES

We conduct additional ablation studies to evaluate the performance differences between D-TRAK and DAS. In this section, CIFAR-2 serves as our primary setting. Further details on these settings

Table 5: We compare our methods with TRAK and D-TRAK by LDS method on CIFAR-2 among different selected timesteps. The projected dimension $k = 4096$.

| Method | Validation | | | Generation | | |
|--------|-------|-------|-------|-------|-------|-------|
|        | 10 | 100 | 1000 | 10 | 100 | 1000 |
| TRAK   | 10.66 | 19.50 | 22.42 | 5.14 | 12.05 | 15.46 |
| D-TRAK | 24.91 | 30.91 | 32.39 | 16.76 | 22.62 | 23.94 |
| DAS    | **33.04** | **42.02** | **43.13** | **20.01** | **29.58** | **30.58** |

Table 6: We compute DAS only with the Up-Block gradients in U-Net and evaluate by LDS method on CIFAR-2 among different selected timesteps. The projected dimension $k = 32768$.

| Method | Validation | | Generation | |
|--------|-------|-------|-------|-------|
|        | 10 | 100 | 10 | 100 |
| D-TRAK | 24.91 | 30.91 | 16.76 | 22.62 |
| DAS(Up-Block) | 32.60 | 37.90 | 18.47 | 27.54 |
| DAS(U-Net) | **33.77** | **42.26** | **21.24** | **29.60** |

are available in Appendix E.1. We establish the corresponding LDS benchmarks as outlined in Appendix E.2.

**Checkpoint selection** Following the approach outlined by Pruthi et al. (2020), we investigated the impact of utilizing different model checkpoints for gradient computation. As depicted in Figures 3, our method achieves the highest LDS when utilizing the final checkpoint. This finding suggests that the later stages of model training provide the most accurate reflections of data influence, aligning gradients more closely with the ultimate model performance. Determining the optimal checkpoint for achieving the best LDS score requires multiple attributions to be computed, which significantly increases the computational expense. Additionally, in many practical scenarios, access may be limited exclusively to the final model checkpoint. This constraint highlights the importance of developing efficient methods that can deliver precise attributions even when earlier checkpoints are not available.

**Value of $\lambda$** In our computation of the inverse of the Hessian matrix within the DAS framework, we incorporate the regularization parameter $\lambda$, as recommended by Hastie (2020), to ensure numerical stability and effective regularization. Traditionally, $\lambda$ is set to a value close to zero; however, in our experiments, a larger $\lambda$ proved necessary. This is because we use the generalized Gauss-Newton (GGN) matrix to approximate the Hessian in the computation of DAS. Unlike the Hessian, the GGN is positive semi-definite (PSD), meaning it does not model negative curvature in any direction. The main issue with negative curvature is that the quadratic model predicts unbounded improvement in the objective when moving in those directions. Without certain techniques, minimizing the quadratic model results in infinitely large updates along these directions. To address this, several methods have been proposed, such as the damping technique discussed in (Martens, 2020). In our paper, we adopt the linear damping technique $\lambda I$ used in (Zheng et al., 2024; Georgiev et al., 2023), which has proven effective on diffusion models. We show how $\lambda$ influence the LDS result in Figure 4, Figure 5, Figure 6, Figure 7 and Figure 8.

## G COUNTER FACTUAL EXPERIMENT

Hu et al. (2024) discuss some limitations of LDS evaluation in data attribution. To further validate the effectiveness of DAS, we also conduct an counter-factual experiment, that 60 generate images are attributed by different attribution method, including TRAK, D-TRAK and DAS. We detect the top-1000 positive influencers identified by these methods and remove them from the training set and re-train the model. We utilize 100 timesteps and a projection dimension of $k = 32768$ to identify the top-1000 influencers for TRAK, D-TRAK and DAS. Additionally, we conduct a baseline experiment where 1000 training images are randomly removed before retraining. The experiment is conducted on ArtBench2 and CIFAR2. We generate the new images with same random seeds and compute

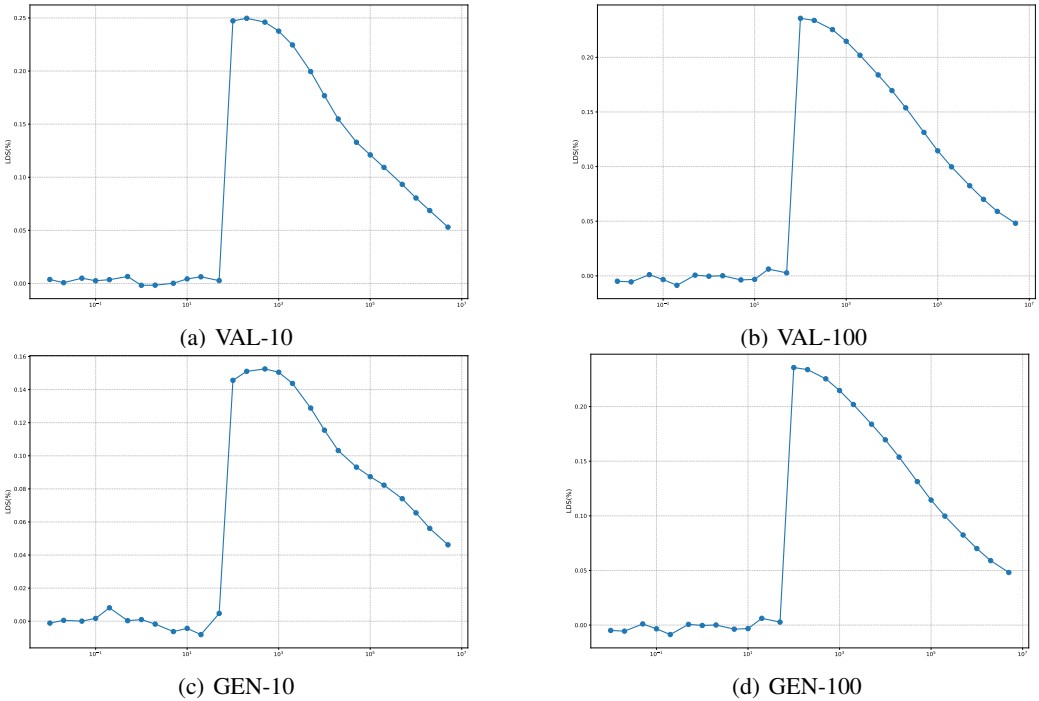

Figure 7: LDS (%) on CIFAR-10 under different $\lambda$. We consider 10 and 100 timesteps selected to be evenly spaced within the interval $[1, T]$, which are used to approximate the expectation $\mathbb{E}_t$. We set $k = 32768$.

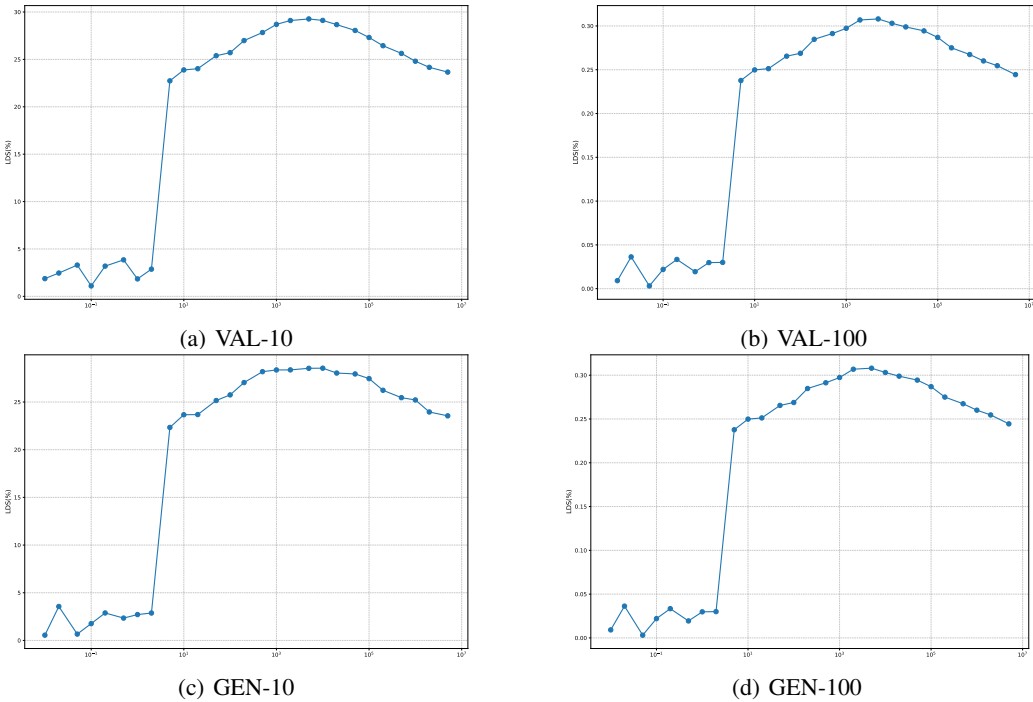

Figure 8: LDS (%) on CelebA under different $\lambda$. We consider 10 and 100 timesteps selected to be evenly spaced within the interval $[1, T]$, which are used to approximate the expectation $\mathbb{E}_t$. We set $k = 32768$.

Table 7: We compute DAS only with the Up-Block gradients in U-Net and evaluate by LDS method on CIFAR-2 among different selected timesteps. The projected dimension $k = 32768$.

| Method | Validation | | Generation | |
|---|---|---|---|---|
| | 10 | 100 | 10 | 100 |
| D-TRAK | 24.91 | 30.91 | 16.76 | 22.62 |
| DAS(Candidate Set) | 31.53 | 37.75 | 17.73 | 23.31 |
| DAS(Entire Training Set) | **33.77** | **42.26** | **21.24** | **29.60** |

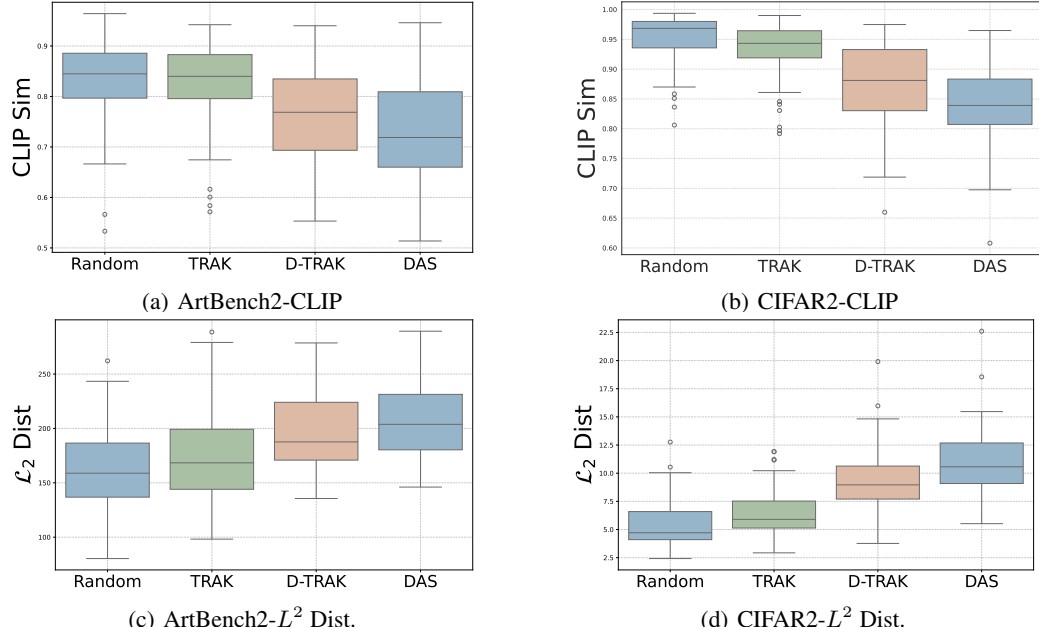

(a) ArtBench2-CLIP  (b) CIFAR2-CLIP

(c) ArtBench2-$L^2$ Dist.  (d) CIFAR2-$L^2$ Dist.

Figure 9: Box-plots of counterfactual evaluation on CIFAR2 and ArtBench2. We assess the impact of removing the 1,000 highest-scoring training samples and retraining the model using Random, TRAK, D-TRAK, and DAS. The evaluation metrics include pixel-wise $L^2$-Distance and CLIP cosine similarity between 60 generated samples and the corresponding images generated by the retrained models, sampled from the same random seed.

the pixel-wise $L^2$-Distance and CLIP cosine similarity between the re-generated images and their corresponding origin images. The result is reported in Figure 9 with boxplot. For the pixel-wise $L^2$-Distance, D-TRAK yields values of 8.97 and 187.61 for CIFAR-2 and ArtBench-2, respectively, compared to TRAK's values of 5.90 and 168.37, while DAS results in values of 10.58 and 203.76. DAS achieves median similarities of 0.83 and 0.71 for ArtBench-2 and CIFAR-2, respectively, which are notably lower than TRAK's values of 0.94 and 0.84, as well as D-TRAK's values of 0.88 and 0.77, demonstrating the effectiveness of our method.

## H  EVALUATING OUTPUT FUNCTION EFFECTIVENESS

Here, we present a toy experiment to validate our theoretical claims: using the Simple Loss Value to represent changes in generated images is inadequate, as it relies on an indirect distributional comparison, as discussed in Eq. 8. Instead, we propose using changes in the noise predictor output of the diffusion model. In the toy experiment, we first train a unconditional DDPM on CIFAR2. The origin and each retrained model are to generate an image pair where one random seed for a re-trained model. In total, we have 60 different generated images pairs. The $L^2$ distance between the generated and original images is calculated to directly measure their differences.

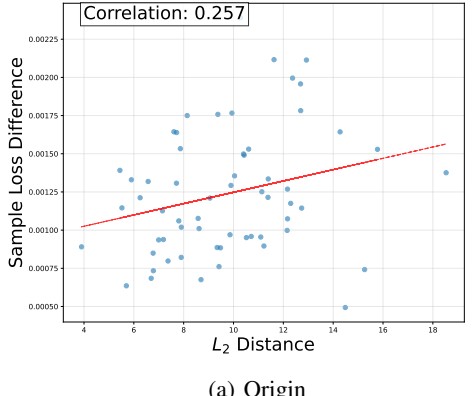
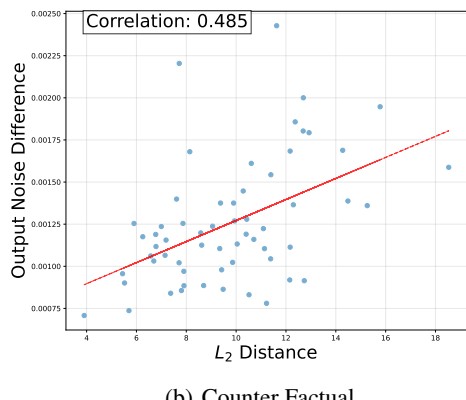

(a) Origin                                (b) Counter Factual

Figure 10: The result of Toy Experiment in Appendix H. Scatter plot showing the relationship between $L^2$ distance and two metrics: loss difference and noise predictor output difference over entire generation. The noise predictor output difference exhibits a stronger correlation with $L^2$ distance, indicating its effectiveness in capturing image variation.

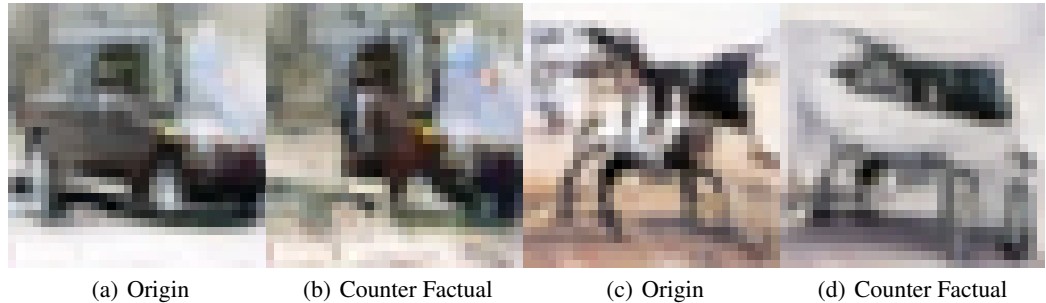

(a) Origin         (b) Counter Factual         (c) Origin         (d) Counter Factual

Figure 11: Figures 11(a) and 11(b) represent a pair of generated images, while 11(c) and 11(d) form another pair. Despite the loss difference between 11(a) and 11(b) being only 0.0007, the $L^2$ distance is 15.261. Similarly, the loss difference between 11(c) and 11(d) is also 0.0007, yet the $L^2$distance is 14.485, showing that the loss value fail to trace the change on generated images.

The original images are first noised to timestep $T$, and the two models are then used to denoise the latent variables at $T$. At the entire denoising process, we compute the average loss difference and the average noise predictor output difference. For the 60 pairs of generated samples, we calculate the rank correlation between these differences and the $L^2$ distance. The results reveal that the Pearson correlation between the $L^2$ distance and the average loss difference is only 0.257, while the correlation with the average noise predictor output difference reaches 0.485. This indicates that the noise predictor output difference aligns better with the $L^2$ distance than the loss value difference. Larger noise predictor output differences correspond to larger $L^2$ distances, reflecting greater image variation. A scatter plot of these results is shown in Figure 10.

Figure 11 provides two specific examples. For images 11(a) and 11(b), as well as 11(c) and 11(d), the loss differences between the models are only 0.0007, the smallest among all pairs. However, the $L^2$ distance between 11(a) and 11(b) is 15.261, and between 11(c) and 11(d) is 14.485, making them among the top five pairs with the largest $L^2$ distances in the toy dataset, which means the loss difference fails to measure the difference on images. Meanwhile, the noise output differences for these two pairs align closely with the trend line in Figure 10(b), showing a correlation with the $L^2$ distance.

This result is unsurprising. Consider an extreme scenario: a model trained on a dataset containing cats and dogs generates an image of a cat. If all cat samples are removed, and the model is retrained,

the new model would likely generate a dog image under the same random seed. In such a case, the loss values for both images might both remain $v$, resulting in a loss difference of 0. This may occurs because the simple loss reflects the model's convergence for the generation, not the image itself. Thus, the shift on loss value fail to capture the extent of image changes. By contrast, analyzing the differences in the noise predictor's outputs at each timestep allows us to effectively trace the diffusion model output on the images.

## I  APPLICATION

Recent research has underscored the effectiveness of data attribution methods in a variety of applications. These include explaining model predictions (Koh & Liang, 2017; Ilyas et al., 2022), debugging model behaviors (Shah et al., 2023), assessing the contributions of training data (Ghorbani & Zou, 2019; Jia et al., 2019), identifying poisoned or mislabeled data (Lin et al., 2022), most influential subset selection (Hu et al., 2024) and managing data curation (Khanna et al., 2019; Liu et al., 2021; Jia et al., 2021). Additionally, the adoption of diffusion models in creative industries, as exemplified by Stable Diffusion and its variants, has grown significantly (Rombach et al., 2022; Zhang et al., 2023). This trend highlights the critical need for fair attribution methods that appropriately acknowledge and compensate artists whose works are utilized in training these models. Such methods are also crucial for addressing related legal and privacy concerns (Carlini et al., 2023; Somepalli et al., 2023).

## J  LIMITATIONS AND BROADER IMPACTS

### J.1  LIMITATIONS

While our proposed Diffusion Attribution Score (DAS) showcases notable improvements in data attribution for diffusion models, several limitations warrant attention. Firstly, although DAS reduces the computational load compared to traditional methods, it still demands significant resources due to the requirement to train multiple models and compute extensive gradients. This poses challenges particularly for large-scale models and expansive datasets. Secondly, the current implementation of DAS is tailored primarily to image generation tasks. Its effectiveness and applicability to other forms of generative models, such as those for generating text or audio, remain untested and may not directly translate. Furthermore, DAS operates under the assumption that the influence of individual training samples is additive. This simplification may not accurately reflect the complex interactions and dependencies that can exist between samples within the training data.

### J.2  BROADER IMPACTS

The advancement of robust data attribution methods like DAS carries substantial ethical and practical implications. By enabling a more transparent linkage between generated outputs and their corresponding training data, DAS enhances the accountability of generative models. Such transparency is crucial in applications involving copyrighted or sensitive materials, where clear attribution supports intellectual property rights and promotes fairness. Nonetheless, the capability to trace back to the data origins also introduces potential privacy risks. It could allow for the identification and extraction of information pertaining to specific training samples, thus raising concerns about the privacy of data contributors. This highlights the necessity for careful handling of data privacy and security in the deployment of attribution techniques. The development of DAS thus contributes positively to the responsible use and governance of generative models, aligning with ethical standards and fostering greater trust in AI technologies. Moving forward, it is imperative to continue exploring these ethical dimensions, particularly the balance between transparency and privacy. Ensuring that advancements in data attribution go hand in hand with stringent privacy safeguards will be essential in maintaining the integrity and trustworthiness of AI systems.

