# OpenReview forum: "Diffusion Attribution Score: Evaluating Training Data Influence in Diffusion Models"
_ICLR.cc/2025/Conference — ICLR 2025 Spotlight_

### Official Review · Reviewer_e8RA · 2024-10-31

**Soundness:** 3
**Presentation:** 3
**Contribution:** 3
**Rating:** 6
**Confidence:** 4

**Summary:**

The paper introduces a new data attribution method called "Diffusion Attribution Score (DAS)" to quantify the influence of training samples in diffusion models, specifically addressing the challenges of data attribution in generative tasks. Experimental results demonstrate that DAS achieves state-of-the-art performance in LDS across various datasets and diffusion models, with strategies explored to accelerate DAS computation for large-scale models.

**Strengths:**

1. Compared to previous work, the proposed DAS method shows significant improvements in experimental results.
2. The article provides anonymized source code, demonstrating the reproducibility of the proposed approach.
3. The data attribution problem in diffusion models studied in this paper is meaningful.

**Weaknesses:**

1. The author mentions "these approaches measure the divergence between predicted and ground truth distributions, which leads to an indirect comparison between the predicted distributions and cannot represent the variances between model behaviors." and provides a brief explanation in Section 3.1. However, as an important argument of the paper, the vague explanation given in lines 178–180 is insufficient to convincingly illustrate the shortcomings of previous work.
2. Similar to W1, lines 191–194 lack adequate support, whether theoretical justification, appropriate citations, or experimental evidence.
3. The logical connection from Sections 3.2 to 3.3 is unclear. The motivation for each sub-module, the problems they aim to solve, and why the proposed methods were chosen are not explained. Additionally, the theoretical proof follows similar derivations as [1] without clearly explaining the motivation behind each step, making it difficult to follow the authors' reasoning.



[1] Park, Sung Min, et al. "TRAK: Attributing Model Behavior at Scale." *International Conference on Machine Learning*. PMLR, 2023.

**Questions:**

Please refer to the Weakness

---

> ### Author Response · Authors · 2024-11-23
>
> Thank you for your valuable review and suggestions. Below we try to respond to the comments in Weaknesses (W) and Questions (Q).
> ***
> # W1: Comparison between Equation (7) and (9)
>
> In data attribution, the objective is to quantify the model distributions shift after the removal of a training sample followed by retraining.
> In this case, we have three distributions, data distribution $q$, original model distribution $p$ and the counterfactual model distribution $p'$.
> In Section 3.1, the shift is measured by computing the KL divergence between the model distributions $p$ and $p'$, as utilized in the DAS method:
> $$
> D_{KL}(p||p')=\int p\log p-p\log p'\ dx
> $$
> This is the most intuitive way to measure the difference between two distributions.
> However, D-TRAK uses the difference between the model distributions and the data distribution, which is represented by KL-divergence between $p$, $p'$ and $q$:
> $$
> D_{KL}(p||q)-D_{KL}(p'||q)=\int p\log - p'\log p'\ dx - \int (p-p')\log q\ dx
> $$
> The computation of D-TRAK involves a term $\int (p-p')\log q\ dx$, which represents the cross-entropy between the distribution shift $p-p'$ and the data distribution $q$.
> This term is generally nonzero unless $p$ and $p'$ are identical distributions.
> By including this term, D-TRAK involves the influence of the data distribution $q$ in the attribution process, whereas DAS isolates the effect of the removed data on the model itself.
> Therefore, DAS assessing the importance of data points based on their actual impact on the model, without confounding influences from the data distribution $q$.
>
> # W2: Comparison between Equation (9) and (10)
>
> Equation (10), derived from Equation (8), aims to empirically eliminate computations involving $q$, instead focusing directly on the change in the model distribution.
> However, a comparison of Equation (10) with Equation (9) reveals that, although Equation (10) removes $q$ from the computation, it does not consider the context in diffusion models.
> Specifically, it treats the model output (noise predictor) as a scalar rather than as a vector in latent space.
> This simplification introduces an error:
> $$
> \tau_{Das}-\tau_{Square}=2E_{\epsilon,t}\langle\epsilon_{\theta_{\backslash i}},\epsilon_{\theta_{\backslash i}}-\epsilon_\theta\rangle
> $$
> $\tau_{Square}$ neglects the high-dimensional information embedded in the model output by omitting the inner product term, which involves cosine similarity information.
> By comparison, the difference between $L_{simple}$ and $L_{DAS}$ is:
> $$
> \tau_{Das}-\tau_{Simple}=2E_{\epsilon,t}\langle\epsilon,\epsilon_\theta-\epsilon_{\theta_{\backslash i}}\rangle + 2E_{\epsilon,t}\langle\epsilon_{\theta_{\backslash i}},\epsilon_{\theta_{\backslash i}}-\epsilon_\theta\rangle$$
> In this case, $\tau_{Simple}$ further includes an additional cross term involving $\epsilon$, which depends on its sampling.
> However, in practice, accurately approximating the expectation of $\epsilon$ can be computationally expensive, leading to potential instability in calculating $\tau_{Simple}$ due to resource limitations.
> ***
> # W3: Logical Connection Improvement
> Thank you for pointing out the unclear part.
> We provide a detailed explanation of the logical flow in Sections 3.2 and 3.3 here.
> The theoretical proof in Section 3.2 is broadly divided into two parts:
> First, we use the Taylor expansion to linearize the model's output function, as shown in Equation (12).
> Then, we evaluate the difference in the output function through the difference in model parameters.
> This evaluation is approximated using Newton's method, as presented in Equation (15), which forms the second part of the proof.
> By combining these two parts, we derive Equation (16).
> However, directly computing Equation (16) is computationally infeasible due to the requirement of inverting the Hessian matrix over the entire training set, which is prohibitively expensive for large-scale models and datasets.
> To address this challenge, we propose several techniques to accelerate the computation of Equation (16) in Section 3.3.
> One key approach is dimensionality reduction for the gradients, simplifying the computation of the inverse.
> Another critical approach is reducing computational overhead.
> The main challenge arises from calculating the gradient for each training sample over sufficient timesteps $t$ and noise $\epsilon$ to approximate the expectation accurately.
> To mitigate this, we reduce both the number of sampled timesteps $t$ and the candidate training samples.
> We have improved the logical flow between Sections 3.2 and 3.3 to clarify these points in the paper.

---

> > ### Comment · Reviewer_e8RA · 2024-11-25
> > **Raising my score**
> >
> > Thanks for the author for the detailed reply, I'm raising my score to 6

---

> > > ### Author Response · Authors · 2024-11-25
> > >
> > > Thanks for raising your score.

---

### Official Review · Reviewer_T3TW · 2024-11-02

**Soundness:** 3
**Presentation:** 3
**Contribution:** 3
**Rating:** 10
**Confidence:** 4

**Summary:**

The authors introduce the Diffusion Attribution Score (DAS) to address limitations found in D-TRAK, a data attribution method for diffusion models. This is a timely and surprisingly under-explored task: While data attribution task (AKA data valuation), which strives to assess training samples' impact on trained model's predictions, has a long history - only recently methods for diffusion models emerged, motivated by the timely copyrights issues that such models present. The previous method D-TRAK approximates comparisons between prediction loss of diffusion models trained with and without specific training samples. However, the authors argue that the reliance on diffusion loss leads to indirect comparisons, and hypothesize that this indirect approach does not accurately capture the influence of training samples on model outputs. To overcome this, they propose DAS, which approximates a direct comparison. As demonstrated in thorough evaluations, this approach indeed provides more precise data attribution scores, when compared to D-TRAK and other baselines.

**Strengths:**

1. The authors picked up on the counterintuitive finding of D-TRAK that switching loss functions in their method, to a loss that is not used during training, increases attribution correctness. Though D-TRAK proposed an explanation, I personally share the author’s belief that this finding calls for further exploration (in my view D-TRAK's explanation requires further substantiation in itself).

2. The authors provide a significant observation: D-TRAK’s attribution scores, as expressed in Eq. (7), do not directly compare (approximate) predictions with/ without training samples. I agree with this found limitation.

3. They propose to derive attribution formulas, that fall within the TRAK methodology, but starting from a direct comparison between the predictions (Eqs. 9). This original paradigm not only solves the newfound limitation, but also sheds light on the counterintuitive finding of D-TRAK: L_square does indeed differ from L_simple by providing a direct comparison of the predictions (Eq. 10).

4. Sufficient performance experiments: The authors demonstrate that DAS indeed outperforms D-TRAK (and several other baselines) across four datasets, and in several settings. Evaluations are in terms of LDS - a widely accepted choice in evaluating TRAK-based methods. Ontop of being appropriate, it is also extensive as it requires multiple training.

**Weaknesses:**

1. The paper lacks a "Previous Work" section—a significant omission, as it currently mentions only one attribution method for diffusion models (D-TRAK) in the introduction. The authors appear to have overlooked other relevant approaches, such as [1], [2], and [3], which should be acknowledged, even if not necessarily included in direct comparisons. Acknowledging these methods is especially important, given that this domain is still underexplored, and the current structure of the paper will give researchers a misleading impression that only gradient-based methods are available. Additionally, key baselines like Journey TRAK, TracInCP and others are introduced in the evaluation tables without prior context - an earlier introduction would clarify their relevance.

2. The mathematical claims lack direct verification through numerical experiments or even qualitative examples, and the improved performance alone is too indirect to serve as convincing validation  **[Edit: i.e. better LDS scores against competitors on standard benchmarks is not enough to substantiate the mathematical claims]**. With that said, given that it is common practice in ML to present mathematical derivations and substantiate them only via improved performance, this is not a critical flaw in terms of acceptance. However, it would significantly improve the impact if such experiments were added - here is a possible idea (other convincing experiments can work as well): Find cases where DAS succeeds and D-TRAK fails, and show that D-TRAK is indeed limited by its indirect comparisons (Sec. 3.1), where DAS solves this.

[1] Wang et. al, ICCV 2023, “Evaluating Data Attribution for Text-to-Image Models”

[2] Brokman et. al, ECCV 2024, “MONTRAGE: Monitoring Training for
Attribution of Generative Diffusion Models”

[3] Kwon et. al, ICLR 2024, “DataInf: Efficiently Estimating Data Influence in LoRA-tuned LLMs and Diffusion Models”

**Questions:**

Please refer to weaknesses

---

> ### Author Response · Authors · 2024-11-23
>
> Thank you for your valuable review and suggestions. Below we try to respond to the comments in Weaknesses (W) and Questions (Q).
> ***
> # W1: Related Work Section Improvement
>
> Thank you for your suggestion.
> Due to page limitations, the Related Work section has been placed in Appendix A.2, with a reference to it added in the Introduction.
> Similarly, when discussing the baseline settings in Section 4.4, we included a reference to the detailed explanation of various attribution methods in Appendix F.3.
> If future page limits allow, we will relocate the Related Work section and the detailed introduction of key baselines to the main body of the paper.
> ***
> # W2: Further validation of DAS
>
> To validate the effectiveness of DAS, we design a counterfactual generation task on ArtBench-2 and CIFAR-2 to demonstrate its ability to identify the most influential samples.
> We compare various attribution methods, including TRAK, D-TRAK, DAS, and Random deletion, as baselines for detecting the 1,000 most influential samples.
> These identified samples are then removed from the training set, and the model is retrained to generate images using the same random seed.
> To evaluate the results, we compute the pixel-wise $L_2$-Distance and CLIP cosine similarity between the re-generated images and their corresponding original images.
> For pixel-wise $L_2$-Distance, DAS achieves values of 10.58 and 203.76 for CIFAR-2 and ArtBench-2, respectively, compared to TRAK's 5.90 and 168.37, while D-TRAK yields 8.97 and 187.61.
> In terms of CLIP cosine similarity, DAS achieves median values of 0.83 and 0.71 for ArtBench-2 and CIFAR-2, respectively, which are lower than TRAK's 0.94 and 0.84 and D-TRAK's 0.88 and 0.77.
> Detailed experimental settings and results are provided in Appendix H.

---

> > ### Comment · Reviewer_T3TW · 2024-11-23
> >
> > W1: It is not usual and borderline unacceptible, especially in practical yet under-explored domains.
> >
> > W2: Please re-read the question. It is not about benchmark evaluations, which are sufficient. It is about empirically demonstrating your mathematical claims, and thus providing empirical insights as to *why* your performance is better than D-TRAK.

---

> ### Author Response · Authors · 2024-11-25
>
> Thanks for your comment.
>
> For the W1, we have modified and improved the paper version to put the related work section in the main paper.
>
> For the W2, we conduct a toy example to empirically demonstrate our mathematical claims: using the Simple Loss Value to represent changes in generated images is inadequate, as it relies on an indirect distributional comparison, as discussed in Eq. (8).
>
> In the toy, we train a unconditional DDPM on CIFAR2, and retrain 60 models after randomly deleting 1,000 samples.
> The origin and each retrained model are to generate an image pair where one random seed for a re-trained model.
> In total, we have 60 different generated images pairs.
> We compute the $L_2$ distance between the pair as the change on the generated data over the re-training process.
> The original images are first noised to timestep $T$, and the two models are then used to denoise the latent variables at $T$.
> We compute the average simple loss difference and the average noise predictor output difference over the entire denise process given by origin and retrained models.
> For the 60 pairs of generated samples, we calculate the rank correlation between these differences and the $L_2$ distance.
>
> The results reveal that the Pearson correlation between the $L_2$ distance and the average loss difference is only 0.257, while the correlation with the average noise predictor output difference reaches 0.485.
> This indicates that the noise predictor output difference aligns better with the $L_2$ distance than the loss value difference.
> Larger noise predictor output differences correspond to larger $L_2$ distances, reflecting greater image variation.
> A scatter plot of these results is shown in Figure 11.
>
> Figure12 provides two detailed examples.
> Figures 12(a) and 12(b) represent one pair of generated images, while 12(c) and 12(d) form another pair.
> For images 12(a) and 12(b), as well as 12(c) and 12(d), the loss differences between the models are only 0.0007, the smallest pairs among all samples.
> However, the $L_2$ distance between 12(a) and 12(b) is 15.261, and between 12(c) and 12(d) is 14.485, making them among the top five pairs with the largest $L_2$ distances in the toy dataset.
> At the same time, the noise output difference for these two pairs lies near the trend line in Figure 11(b).
>
> This result is unsurprising.
> Consider an extreme case: a model trained on a dataset containing cats and dogs generates an image of a cat.
> If all cat samples are removed, and the model is retrained, the new model would likely generate a dog image under the same random seed.
> In this case, the loss values for origin and counter-factual images might both be $v$, resulting in a loss difference of 0.
> This may occurs because the simple loss reflects the model’s convergence for a generation, not the generated image itself.
> Thus, loss differences fail to capture the extent of image changes.
> By contrast, analyzing the differences in the noise predictors' outputs at each timestep allows us to effectively trace the diffusion model output on the images.

---

> ### Comment · Reviewer_T3TW · 2024-11-25
>
> Thank you for the improvements and clarifications. Since both my concerns were resolved, I raise my score from 6 to 10. The studied task, though hard to solve, has immense implications, and it is not studied enough, even though achieving reliable technology would transform legal and economical aspect of digital art. The advancements here should be highlighted, since your work clearly solves a fundamental mathematical issue in D-TRAK, one of the important directions of this domain. Best of luck.

---

### Official Review · Reviewer_AfWj · 2024-11-04

**Soundness:** 2
**Presentation:** 2
**Contribution:** 2
**Rating:** 6
**Confidence:** 3

**Summary:**

This paper introduces a novel metric called Diffusion Attribution Score (DAS) to more accurately quantify the attribution of training samples to generated images. While the paper is tackling an important problem of accurate data attribution for diffusion models, it has areas for improvement, such as justification of their proposed and more rigorous evaluation of the method as described below.

**Strengths:**

- The paper addresses the important problem of attributing diffusion models’ output to training samples.
- The metrics in quantitative evaluation look good, all showing significant gaps accounting for the confidence interval or standard deviation (though further clarity on these intervals would be helpful as they are not noted).

**Weaknesses:**

- The equations in Section 3.1 only show that DAS is independent of data distribution, unlike D-TRAK. However, it has not been shown how significant it is. While DAS seems to be working well in the experiment results, the theory behind it appears weak.
- The evaluation solely relies on the LDS metric, which is based on the strong assumption of additive data contributions. There have been discussions on the effectiveness of LDS for evaluating data attribution algorithms, such as https://arxiv.org/abs/2409.18153 Adding other metrics or justifying its use looks necessary.
- In Figure 1, diffusion model outputs from the same random seeds are presented. The rationale behind the comparison seems to be that the same random seed should lead to a similar output when models are trained on datasets of similar distribution. However, this may not be true since models are retrained, and they are essentially different ones. How can you ensure that two diffusion models trained on the same dataset generate the same output for a random seed when other randomness, such as batch sampling, is not controlled?

**Questions:**

- Could you please clarify the theoretical benefits of DAS over D-TRAK?

---

> ### Author Response · Authors · 2024-11-23
>
> Thank you for your valuable review and suggestions. Below we try to respond to the comments in Weaknesses (W) and Questions (Q).
> ***
> # W1, Q1: Comparison between DAS and D-TRAK
>
> In data attribution, the objective is to quantify the model distributions shift after the removal of a training sample followed by retraining.
> In this case, we have three distributions, data distribution $q$, original model distribution $p$ and the counterfactual model distribution $p'$.
> In Section 3.1, the shift is measured by computing the KL divergence between the model distributions $p$ and $p'$, as utilized in the DAS method:
> $$
> D_{KL}(p||p')=\int p\log p-p\log p'\ dx
> $$
> This is the most intuitive way to measure the difference between two distributions.
> However, D-TRAK uses the difference between the model distributions and the data distribution, which is represented by KL-divergence between $p$, $p'$ and $q$:
> $$
> D_{KL}(p||q)-D_{KL}(p'||q)=\int p\log - p'\log p'\ dx - \int (p-p')\log q\ dx
> $$
> The computation of D-TRAK involves a term $\int (p-p')\log q\ dx$, which represents the cross-entropy between the distribution shift $p-p'$ and the data distribution $q$.
> This term is generally nonzero unless $p$ and $p'$ are identical distributions.
> By including this term, D-TRAK involves the influence of the data distribution $q$ in the attribution process, whereas DAS isolates the effect of the removed data on the model itself.
> Therefore, DAS assessing the importance of data points based on their actual impact on the model, without confounding influences from the data distribution $q$.
> ***
> # W2: Counterfactual Generation
>
> [1] provides an insightful discussion on the limitations of LDS, and we elaborate on the contributions of [1] and the limitations of LDS in the related work section.
>
> To further validate the effectiveness of DAS, we construct a counterfactual generation task on ArtBench-2 and CIFAR-2 to demonstrate that DAS can effectively identify the most influential samples.
> We use various attribution methods, including TRAK, D-TRAK, DAS, and Random deletion as baselines, to detect the 1,000 most influential samples and then remove them from the training set.
> The model is retrained and generate the images with the same random seed.
> We compute the pixel-wise $l_2$-Distance and CLIP cosine similarity between the re-generated images and their corresponding images.
>
> For the pixel-wise $L_2$-Distance, DAS achieves values of 10.58 and 203.76 for CIFAR-2 and ArtBench-2, respectively, compared to TRAK's values of 5.90 and 168.37, while D-TRAK yields values of 8.97 and 187.61.
> In terms of CLIP cosine similarity, DAS achieves median values of 0.83 and 0.71 for ArtBench-2 and CIFAR-2, respectively, which are lower than TRAK's values of 0.94 and 0.84 and D-TRAK's values of 0.88 and 0.77.
>
> Details of the experiment and result are provided in Appendix H.
>
> [1] Hu et al. (2024) - Most Influential Subset Selection: Challenges, Promises, and Beyond.
> ***
> # W3: Other randomness shifts in retraining
>
> A model is relied on various factors, such as the training set, model architecture, learning algorithm and training parameters.
> In data attribution, the goal is to determine the importance of a particular training sample on the model's performance for a given test sample, which is achieved by answering a counterfactual question.
> Thus, we fix all the randomness, remove a single training sample from the training set, and then retrain the model.
> It is true that we cannot isolate the effect of a specific training sample because removing a training sample will inevitably affect other randomness, such as batch sampling.
> If we focus only on changes before and after removing a training sample, these effects will also be influenced by some other randomness.
> However, in experiments, when comparing the attribution results, this effect remains consistent.
> For example, when comparing different attribution methods for attributing a specific training sample, or comparing the impact of different training samples on a test sample with a method, the error caused by other randomness shifts is consistent among different methods or training samples.
> Therefore, these comparisons are still valid.

---

> ### Author Response · Authors · 2024-11-25
>
> Respected reviewer, should you have any further concerns, I am eagerly anticipating your response.

---

> > ### Comment · Reviewer_AfWj · 2024-11-25
> >
> > Thank you for your response and additional experiment. I raised my rating.

---

### Official Review · Reviewer_bWtE · 2024-11-04

**Soundness:** 2
**Presentation:** 2
**Contribution:** 2
**Rating:** 8
**Confidence:** 4

**Summary:**

(1) This paper examines the choice of the model output function in D-TRAK, demonstrates the theoretical shortcomings of D-TRAK's choice, and proposes using the predicted noise of a diffusion model as the output function (i.e., the output of the noise predictor). (2) With the proposed model output function, the leave-one-out attribution is approximated with Taylor expansion and Newton's method. (3) Additional tricks such as moving the expectation w.r.t the timestep inside the norm and using CLIP score to select candidate important data are proposed. Finally, the proposed method (i.e., DAS) achieves better linear datamodeling scores than existing attribution methods.

**Strengths:**

- This paper addresses the gap left by Zheng et al. (2024) and demonstrates why the simple diffusion loss is not an appropriate model output function for TRAK-based methods (Equation 8).
- The proposed, theoretically more principled method DAS actually leads to improved linear datamodeling scores.
- This paper proposes to reduce the number of candidate training samples for computing data attribution scores, using CLIP score as the criterion. This approach can significantly improve computational efficiency and is shown to work fine (Table 7).

References
- Zheng et al. (2024) - Intriguing properties of data attribution on diffusion models.

**Weaknesses:**

- Although the paper shows that $L_{\text{simple}}$ is not an appropriate model output function, it is not directly shown why $L_{\text{Square}}$, as proposed in Zheng et al. (2024), is more appropriate. Specifically, Equations (9) and (10) are not exactly equivalent. In this sense, this paper does not fully explain the intriguing findings in Zheng et al. (2024). A more direct theoretical comparison between Equations (9) and (10) would make the contribution greater.

- In lines 153-155 and Equation (7), the paper states that D-TRAK uses the simple loss $L_{\text{simple}}$ as the output function. However, the main experimental results from Zheng et al. (2024) come from D-TRAK implemented with $L_{\text{Square}}$ as the output. This mismatch in writing needs to be resolved.

- Quantitative results are missing in the experiment where the top 1,000 positive influential samples are removed. Something like the counterfactual evaluation in Zheng et al. (2024) would suffice.

References
- Zheng et al. (2024) - Intriguing properties of data attribution on diffusion models.

**Questions:**

- How does Equation (9) validate the effectiveness of Equation (10) in comparison to the ineffectiveness of Equation (7)?
- Why do you think $\lambda$ has such a big impact on the performance of DAS (shown in Figure 4)? How is $\lambda$ tuned for CelebA and ArtBench?
- In the proof for applying Newton's Method, it is assumed that $\theta^*$ is a global optimum. Can you explain why this assumption is reasonable for diffusion models?
- Since Newton's Method can be applied to approximate $\theta^*_{\setminus i}$, why can't we used the approximated $\theta^*_{\setminus i}$ in Equation (9)? This way we don't need to linearize $f_{\text{DAS}}$.

---

> ### Author Response · Authors · 2024-11-23
>
> Thank you for your valuable review and suggestions. Below we try to respond to the comments in Weaknesses (W) and Questions (Q).
> ***
> # W1,Q1: Comparison between $\tau_{Simple}$, $\tau_{Square}$ and $\tau_{DAS}$
>
> We are pleased to present a detailed theoretical comparison among Equations (7), (9), and (10).
> The objective of data attribution is to quantify the model distributions shift after the removal of a training sample followed by retraining.
> This shift is measured by computing the KL divergence between the model distributions $p$ and $p'$, as utilized in the DAS method:
> $$D_{KL}(p||p')=\int p\log p-p\log p'\ dx$$
> This approach provides the most intuitive way of evaluating the difference between two distributions.
> In contrast, D-TRAK measures the difference between the model distributions and the data distribution $q$, expressed as the KL divergence between $p$, $p'$ and $q$, respectively:
> $$
> D_{KL}(p||q)-D_{KL}(p'||q)=\int p\log - p'\log p'\ dx - \int (p-p')\log q\ dx
> $$
> The computation of D-TRAK involves a term $\int (p-p')\log q\ dx$, which represents the cross-entropy between the distribution shift $p-p'$ and the data distribution $q$.
> This term is generally nonzero unless $p$ and $p'$ are identical distributions.
> By including this term, D-TRAK involves the influence of the data distribution $q$ in the attribution process, whereas DAS isolates the effect of the removed data on the model itself.
>
> Equation (10), derived from Equation (8), aims to empirically eliminate computations involving $q$, instead focusing directly on the change in the model distribution.
> However, a comparison of Equation (10) with Equation (9) reveals that, although Equation (10) removes $q$ from the computation, it does not consider the context in diffusion models.
> Specifically, it treats the model output (noise predictor) as a scalar rather than as a vector in latent space.
> This simplification introduces an error:
> $$
> \tau_{Das}-\tau_{Square}=2E_{\epsilon,t}\langle\epsilon_{\theta_{\backslash i}},\epsilon_{\theta_{\backslash i}}-\epsilon_\theta\rangle
> $$
> $\tau_{Square}$ neglects the high-dimensional information embedded in the model output by omitting the inner product term, which involves cosine similarity information.
> By comparison, the difference between $L_{simple}$ and $L_{DAS}$ is:
> $$
> \tau_{Das}-\tau_{Simple}=2E_{\epsilon,t}\langle\epsilon,\epsilon_\theta-\epsilon_{\theta_{\backslash i}}\rangle + 2E_{\epsilon,t}\langle\epsilon_{\theta_{\backslash i}},\epsilon_{\theta_{\backslash i}}-\epsilon_\theta\rangle$$
> In this case, $\tau_{Simple}$ further includes an additional cross term involving $\epsilon$, which depends on its sampling.
> However, in practice, accurately approximating the expectation of $\epsilon$ can be computationally expensive, leading to potential instability in calculating $\tau_{Simple}$ due to resource limitations.
>
> ***
> # W2: Notation alignment with D-TRAK
>
> This notation is consistent with prior work [1].
> In D-TRAK, the authors initially define the output function as $L_{Simple}$, with the corresponding attribution method represented by $\tau_{D-TRAK}$.
> Subsequently, they propose replacing $L_{Simple}$ with $L_{Square}$ and other output functions to improve attribution performance.
> In this paper, we refer to the attribution method as $\tau_{Square}$ ​when $L_{Square}$ is used as the output function.
>
> ***
> # W3: Quantitative results of counter factual generation
> To further validate the effectiveness of DAS, we present the quantitative results of counterfactual generation in Appendix H, comparing DAS with D-TRAK, TRAK, and Random deletion.
> We use different metrics to evaluate the top 1,000 positive influential samples for 60 generated images on CIFAR-2 and ArtBench-2.
> After removing them from the training set and re-training the model, we generate the images with the same random seed.
> We compute the pixel-wise $l_2$-Distance and CLIP cosine similarity between the re-generated images and their corresponding images.
> For the pixel-wise $L_2$-Distance, DAS achieves values of 10.58 and 203.76 for CIFAR-2 and ArtBench-2, respectively, compared to TRAK's values of 5.90 and 168.37, while D-TRAK yields values of 8.97 and 187.61.
> In terms of CLIP cosine similarity, DAS achieves median values of 0.83 and 0.71 for ArtBench-2 and CIFAR-2, respectively, which are lower than TRAK's values of 0.94 and 0.84 and D-TRAK's values of 0.88 and 0.77.

---

> > ### Author Response · Authors · 2024-11-23
> >
> > # Q2: Impact of $\lambda$
> > In our method, we reduce the computational cost of calculating the Hessian matrix by using the generalized Gauss-Newton (GGN) matrix as an approximation, as described in Equation (23).
> > We need to ensure that the GGN matrix is positive semi-definite (PSD), meaning it does not capture negative curvature in any direction.
> > Negative curvature poses a significant challenge, as the quadratic model can predict unbounded improvement in the objective function when moving along such directions.
> > Without proper techniques, minimizing the quadratic model may result in infinitely large updates in these directions, leading to instability in the optimization process.
> > To address this issue, various methods have been proposed, including the damping technique discussed in [1], to guarantee the GGN is a PSD matrix.
> > In our paper, we adopt the linear damping technique, $\lambda I$, as used in [2][3], which has been demonstrated to be effective for diffusion models.
> > We further evaluate the impact of the damping parameter $\lambda$ on the performance of DAS in the ablation study, with results provided in Appendix G for the ArtBench2, ArtBench5, CIFAR10, and CelebA datasets.
> >
> > [1] Martens J. New insights and perspectives on the natural gradient method[J]. Journal of Machine Learning Research, 2020, 21(146): 1-76.
> >
> > [2] Zheng X, Pang T, Du C, et al. Intriguing properties of data attribution on diffusion models[J]. arXiv preprint arXiv:2311.00500, 2023.
> >
> > [3] Georgiev K, Vendrow J, Salman H, et al. The journey, not the destination: How data guides diffusion models[J]. arXiv preprint arXiv:2312.06205, 2023.
> > ***
> > # Q3: Newton's Method Discussion
> >
> > In the theoretical section, $\theta^*$ is defined as the model parameter at convergence; this is a definition, not an assumption.
> > When the model converges, the parameters remain unchanged, ensuring the validity of Equation (22).
> > In practice, due to the limitations of model optimization, achieving fully converged parameters may not always be feasible, and the model might oscillate near a local optimum.
> > As a result, the assumption underlying Equation (22) may not hold precisely, leading to a small error.
> > However, this has no impact on subsequent DAS calculations, as Equation (22) is not directly involved in the computation of DAS.
> > Any minor error does not undermine the theoretical foundation of DAS.
> > ***
> > # Q4: Applying Newton's Method in Equation (9)
> > Newton's method is used to evaluate changes in model parameters.
> > However, in data attribution, the goal is to assess the importance of a specific data point $i$ on a given test sample.
> > This importance is measured by the change in the model's output for the test sample before and after modifying the training data.
> > Evaluating data attribution cannot rely solely on the change in model parameters; it is equally important to consider how these parameter changes influence the output function.
> > This can be understood abstractly by interpreting Equation (9) using a linear model $Y=\theta^TX$.
> > In Equation (9), the focus is on evaluating changes in the model's output function, that is $Y$ and $Y'$ after removing a training sample.
> > Using Newton's iteration, we can just compute the difference between the model parameters $\theta$ and $\theta_{\setminus i}$ after removing a training sample.
> > We still need to calculate the output difference by $(\theta-\theta_{\setminus i})X$ on the test sample.
> > Thus, linearizing the output function remains essential, as it establishes a connection between changes in the output function and changes in the model parameters.
> > Therefore, the difference on the noise predictor can be measured by the model parameters difference computed by Newton's Method.

---

> ### Author Response · Authors · 2024-11-25
>
> Respected reviewer, should you have any further concerns, I am eagerly anticipating your response.

---

> > ### Comment · Reviewer_bWtE · 2024-12-03
> >
> > The authors' responses to Weaknesses 1 & 3 (W1 & W3) and Question 1 (Q1) have addressed my main concerns. Therefore I raised my score from a 5 to an 8, especially because the analysis on the differences between $\tau_{\text{DAS}}, \tau_{\text{Square}}, \tau_{\text{Simple}}$ is insightful. My remaining comments are minor and don't affect my rating of the paper, but I expect the authors to address them thoughtfully.
> >
> > **W2:** The main contribution of D-TRAK is to show that $L_{\text{Square}}$ is a better output function than $L_{\text{Simple}}$, and the authors of D-TRAK explicitly mention that D-TRAK is defined using $L_{\text{Square}}$ at the beginning of Section 4 of their paper. I suggest the authors respect this definition and modify their wording in lines 203-204.
> >
> > **Q2:** The ablation study seems to suggest that $\lambda$ is chosen around or at the optimal LDS, requiring access to a set of retrained models. Having a set of retrained models may not be practical for large-scale model training (e.g., full Stable Diffusion model training), limiting the applicability of DAS. This is especially concerning because the value of $\lambda$ can have a huge impact on the LDS performance (i.e., choosing the wrong $\lambda$ can result in LDS around 0, as shown in Figure 4). However, I recognize that this is a general limitation of TRAK-based methods for diffusion models, but the authors should include a brief discussion about this issue.
> >
> > **Q3:** Through that definition, you implicitly assume that the model is trained to convergence. Since your estimation result does not rely on the convergence assumption, it would be clearer if you just drop Equation (22).
> >
> > **Q4:** Using Newton's method gives us an estimated $\theta_{\setminus i}$ (by rearranging terms in Equation 25), so we have $\epsilon_{\theta_{\setminus i}}$. Then why can't we plug the $\epsilon_{\theta_{\setminus i}}$ estimated via Newton's method into Equation (9)? Isn't this still feasible although computationally more expensive than linearizing the output function?

---

> ### Author Response · Authors · 2024-12-03
>
> Thank you for acknowledging our theoretical comparison of $\tau_{\text{DAS}}$ with $\tau_{\text{Simple}}$ and $\tau_{\text{Square}}$, and for increasing your score as a result.
> The minor questions you raised are fixable, and we will improve them in future revisions of the paper:
>
> W2: We will revise the notation for the output function $f_{D-TRAK}$ and attribution score $\tau_{D-TRAK}$ to $f_{Simple}$ and $\tau_{Simple}$, respectively, for better clarity regarding the use of "Simple Loss" and "Square Loss."
>
> Q2: As you noted, this is a general limitation of attribution methods that simplify computations by using the Fisher Information Matrix or the generalized Gauss-Newton matrix in place of the Hessian Matrix. We will include a discussion about $\lambda$ in the paper.
>
> Q3: We will refine the notation for the convergence assumption and delete Eq.(22) to enhance clarity.
>
> Q4: Since obtaining $\epsilon_{\theta_{\setminus i}}$ using $\theta_{\setminus i}$ requires linearizing $\epsilon_{\theta_{\setminus i}}$ ((as the diffusion model's output $\epsilon_{\theta_{\setminus i}}$  is a nonlinear function with respect to $\theta_{\setminus i}$), we still cannot avoid linearization of the model output function.

---

### Author Response · Authors · 2024-11-28
**Submission of Revised Manuscript**

Dear Reviewers,

We sincerely thank all the reviewers for their constructive and insightful comments on our work. We have addressed the revisions committed to during the review process, with the main updates as follows:
1. We conducted a detailed comparison and analysis of the model output function design.
2. We have revised the structure of the paper, including the Related Work and Method sections, to improve readability.
3. Additional experiments have been included to provide further experimental evidence supporting the effectiveness of DAS.

Once again, we extend our heartfelt thanks to all reviewers for their time and effort during the review process.

Best,

Authors

---

### Meta-Review · Area_Chair_JeAn · 2024-12-22

**Metareview:**

This paper proposes the diffusion attribution score, which measures the direct comparison between predicted distributions and analyzes the training sample importance. The authors claim that the proposed method is better than existing ones that measure the divergence between predicted and groundtruth distributions. The proposed method is verified through both theoretical analysis and empirical evaluations. In terms of linear data-modeling score, the method achieves the state-of-the-art performance on existing benchmarks. This is a strong paper, and I recommend it for acceptance.

**Additional Comments On Reviewer Discussion:**

The reviewers’ concerns center around 1) the theoretical justification and comparison between simple, squared and the proposed output functions are not fully convincing, 2) the experimental results lack diversity, such as fixed random seeds, potential limitation of the LDS score in evaluation. The authors addressed these concerns in the rebuttal, and one reviewer (AfWj) raised their rating to accept. After rebuttal and discussion, all reviewers give positive ratings.

---

### Decision · Program_Chairs · 2025-01-22

Accept (Spotlight)